# How Rh surface breaks CO$_2$ molecules under ambient pressure

Jeongjin Kim [1,8], Hyunwoo Ha [2,8], Won Hui Doh[1,8], Kohei Ueda[3], Kazuhiko Mase[4], Hiroshi Kondoh[3], Bongjin Simon Mun [5,6 ✉], Hyun You Kim [2 ✉] & Jeong Young Park [1,7 ✉]

Utilization of carbon dioxide (CO$_2$) molecules leads to increased interest in the sustainable synthesis of methane (CH$_4$) or methanol (CH$_3$OH). The representative reaction intermediate consisting of a carbonyl or formate group determines yields of the fuel source during catalytic reactions. However, their selective initial surface reaction processes have been assumed without a fundamental understanding at the molecular level. Here, we report direct observations of spontaneous CO$_2$ dissociation over the model rhodium (Rh) catalyst at 0.1 mbar CO$_2$. The linear geometry of CO$_2$ gas molecules turns into a chemically active bent-structure at the interface, which allows non-uniform charge transfers between chemisorbed CO$_2$ and surface Rh atoms. By combining scanning tunneling microscopy, X-ray photoelectron spectroscopy at near-ambient pressure, and computational calculations, we reveal strong evidence for chemical bond cleavage of O–CO* with ordered intermediates structure formation of (2 × 2)-CO on an atomically flat Rh(111) surface at room temperature.

[1] Center for Nanomaterials and Chemical Reactions, Institute for Basic Science (IBS), Daejeon 34141, Republic of Korea. [2] Department of Materials Science and Engineering, Chungnam National University, Daejeon 34134, Republic of Korea. [3] Department of Chemistry, Keio University, 3-14-1 Hiyoshi, Kohoku-ku, Yokohama 223-8522, Japan. [4] Institute of Materials Structure Science, High Energy Accelerator Research Organization, SOKENDAI (The Graduate University for Advanced Studies), 1-1 Oho, Tsukuba 305-0801, Japan. [5] Department of Physics and Photon Science, School of Physics and Chemistry, Gwangju Institute of Science and Technology (GIST), Gwangju 61005, Republic of Korea. [6] Center for Advanced X-ray Science, GIST, Gwangju 61005, Republic of Korea. [7] Department of Chemistry, Korea Advanced Institute of Science and Technology (KAIST), Daejeon 34141, Republic of Korea. [8] These authors contributed equally: Jeongjin Kim, Hyunwoo Ha, Won Hui Doh. ✉email: bsmun@gist.ac.kr; kimhy@cnu.ac.kr; jeongypark@kaist.ac.kr

Sharply rising carbon dioxide ($CO_2$) levels in the air have escalated a climate crisis that affects the living conditions of humankind, leading to the most uncertain natural environment on our planet since the agricultural revolution[1,2]. Carbon capture, utilization, and storage (CCUS) is a widely accepted energy recycling strategy of lowering $CO_2$ emissions to protect the environment in sustainable industries[3]. Accordingly, the utilization of $CO_2$ for primary energy source production is becoming a central issue in relation to future technologies for renewable energy conversion. In particular, methanation ($CO_2 + 4H_2 \rightarrow CH_4 + 2H_2O$) or methanol synthesis ($CO_2 + 3H_2 \rightarrow CH_3OH + H_2O$) via $CO_2$ reduction reaction ($CO_2RR$) is an important catalytic conversion process for renewable utilization, in that economically useful energy sources could be produced from a primitive single molecule[4,5]. However, strongly binding carbon-to-oxygen intramolecular bonds ($O = C = O$) and their almost nonreactive linear geometry create a challenging drawback for utilization. According to the Gibbs−Helmholtz relationship, the associated enthalpy required to break the intramolecular bonding of $CO_2$ into CO and O is $\Delta H^0 = + 293.0$ kJ/mol, which makes the $CO_2$ molecule very stable[6,7]. Hence, driving several transient intermediates at gas−solid interface facilitates essential reaction steps by inducing the interaction of 2-12 electrons to produce the energy source in heterogenous catalysis[8,9].

One of the primary $CO_2RR$ processes in the industry is $CO_2$ hydrogenation over zinc oxide-supported copper (Cu/ZnO) catalysts for methanol synthesis[10]. This reaction is actively facilitated by the formation of transient formate ($HCOO^-$) species from the chemisorbed $CO_2^{\delta-}$ in harsh conditions (e.g., 50−100 bar and 500−550 K), which demonstrates the critical role of intermediate formation in bypassing the out-of-range reaction routes of favorable thermodynamics[11]. In contrast, the similar feedstock of $H_2$ and $CO_2$ over silicon[12] or aluminum[13] oxide-supported rhodium (Rh) catalyst selectively yields a major product of methane via the conversion of $CO_2$ to CO at an early step in the $CO_2RR$ process[14]. In this way, although various derivatives of the reactant $CO_2$ molecules could affect multiple steps of transition states and thermodynamic equilibrium potentials, as predicted by theoretical calculations on the tailored surfaces[15], previous studies have nonetheless assumed the adsorbate $CO_2$ interactions with insufficient knowledge of underlying molecular behavior in operation conditions. For example, the calorimetry analysis results of $CO_2$ over Rh surfaces suggested controversial conclusions[16,17] about $CO_2$ dissociation between $10^{-9}$ and 1 bar of pressure; they indicated that experimental detections of the dissociated carbon monoxide (CO*; an asterisk means as adsorbed species) and oxygen (O*) are impossible at lower pressures due to the probability of rare dissociative $CO_2$ adsorption in the order of $10^{-15}$. In other words, the unreactive signature molecule only has a chance of catalytic activation at higher pressures, and the presence of the pressure gap issue[18] should be mainly considered for investigating chemical reaction pathways on the surface. However, their specific characterizations have been comparatively limited because of technical difficulties related to the decrease of the electron mean-free path at elevated pressures[19]. Eventually, the progressive tools operable in realistic conditions[20,21] are necessarily required to unravel the early steps of $CO_2RR$ at the molecular level. Recently published literature supports the strong evidence of $CO_2$ activation beyond the pressure gap, for instance, the intramolecular bond-breakage phenomenon of $CO_2$ molecules was reported on the Cu[22,23] and Ni[24,25] catalysts at the elevated pressures.

Here, we report direct observations of $CO_2$ molecules' dissociative adsorption structures at $CO_2$−Rh(111) interface, employing advanced surface science techniques of scanning tunneling microscopy (STM) and X-ray photoelectron spectroscopy (XPS) at near-ambient pressures (NAP). Unlike the physisorption process, the chemisorption of $CO_2$ molecules induces changes of local charge density at the interface in order for the stimulated electronic interactions making a sufficient influence of bond cleavage for adsorbate O−CO*, even at 300 K. Direct imaging results under 0.1 mbar of $CO_2(g)$ reveal that the dissociated O* and CO* occupy the hollow and atop site of each on the atomically flat Rh surface, which brings the spontaneous ordering formation of $(2 \times 2)$-CO adsorption structures with a coverage ($\theta_{CO}$) of 0.25 monolayer (ML) at equilibrium. Time-lapse NAP-XPS measurements also highlight the only significant increase of the dissociated CO* species at the atop site of the Rh surface in the chemical binding analysis of C 1$s$ core-level spectra as well. Density functional theory (DFT) calculations elucidate a possible reaction route for the observed $CO_2$ dissociation process with an activation energy barrier ($E_a$) of 0.58 eV on account of the nonuniform distribution of charge transfers between the bent (b)-$CO_2$ molecule and Rh surface atoms.

## Results

**Molecular adsorption structure observations of CO and $CO_2$.** The well-defined {111}-facet model of the Rh single crystal was prepared in ultra-high vacuum (UHV) before direct observation under CO or $CO_2$ environment. Figure 1a shows the freshly prepared step-terrace structures which have uniformly measured monatomic step heights of 2.2 Å along with dark and bright colors of separate local areas on the pretreated Rh(111) surface. This clean surface is consistent with a theoretical model of Rh surface structure and with literature values from real-space observations using STM[26,27]. Under the $CO_2$ environment, as illustrated in Fig. 1b, linear (l)-$CO_2(g)$ molecules randomly collide with surface Rh atoms with kinetic energy (KE) at a given temperature in statistical velocity distribution; such fundamental molecular motions are correlated to a simplified function of chemical potential energy in the established ideal solution at equilibrium. With the right amount of $CO_2$, the ground state of l-$CO_2(g)$ could have a greater chance for electronic transition throughout the excited state by exchanging hot carriers at the gas−solid interface[28,29], and nonequilibrium molecular behaviors may result in the geometric transformation to b-$CO_2$ (ads.) on the Rh surface[30,31].

Representative atom-resolved resolution of NAP-STM images in Fig. 1c−e show the different chemisorption properties of CO and $CO_2$ molecules on the Rh(111) surface. The observed hexagonal pattern of Rh atoms in UHV is periodically arrayed with a nearest-neighborhood distance of 2.7 Å in Fig. 1c, corresponding to |a| of the illustrated atomistic ball model in Fig. 1f. The clean Rh terraces are immediately covered by introduced CO molecules in a 0.1 mbar CO environment; the ordered bright spots and mismatched discrete lines appear simultaneously, as shown in Fig. 1d. A unit cell of the observed structural orderings by CO/Rh is indicated as a dotted parallelogram, which has long-range patterns attributed to $(2 \times 2)$-3CO structures of 0.75 ML, as illustrated in Fig. 1g. The model construction of CO/Rh structure explains that each corner of a parallelogram is connected to an atop (t) site of CO* with a length of 2|a| on each side, which meets at an angle of 60° per unit cell. Also, two CO molecules occupy the hollow (h) sites inside the parallelogram, and the proposed CO/Rh structure at $\theta_{CO}$ of 0.75 ML consists of t-CO and h-CO with a ratio of 1:2 on the Rh (111) surface[32,33]. However, the displayed NAP-STM image under 0.1 mbar CO environment is not perfectly matched with the proposed model structure, due to changes of tunneling current probability in measurement condition. In principle, the local electron densities of t-CO and h-CO have different bond

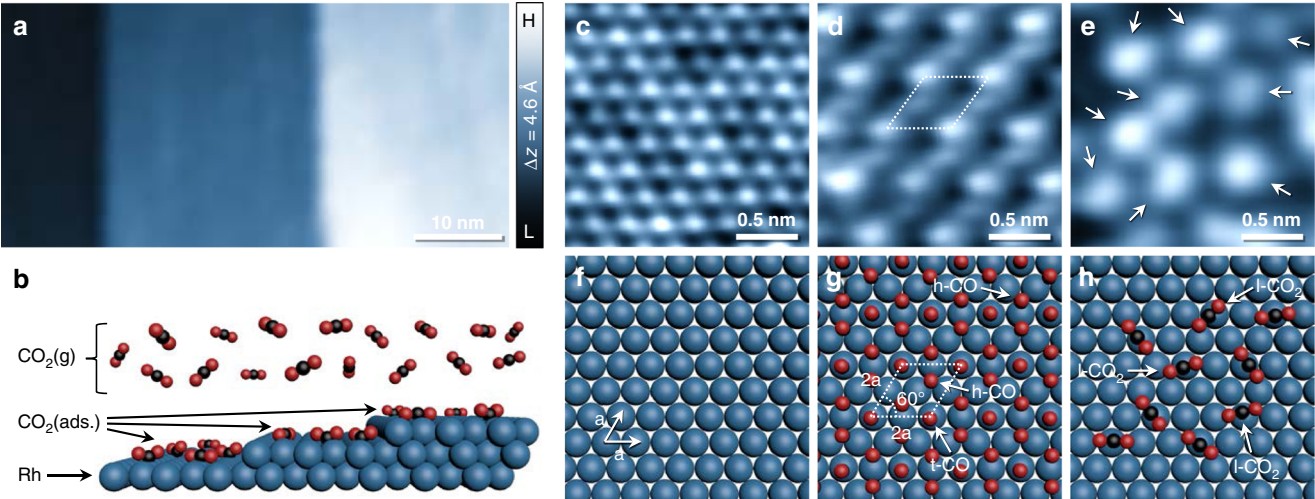

**Fig. 1 Molecular adsorption structure of CO and $CO_2$ on the Rh(111) surface. a** Freshly prepared wide-range STM image in UHV after cleaning cycles ($V_t$ = 1.25 V; $I_t$ = 0.17 nA). **b** Schematic illustration of the gaseous $CO_2$/Rh(111) interface (side view). **c–e** Representative atomic-resolution NAP-STM images of clean in UHV ($V_t$ = 0.23 V; $I_t$ = 0.25 nA) (**c**), 0.75 ML of (2 × 2)-3CO chemisorption structure at 0.1 mbar CO ($V_t$ = 1.03 V; $I_t$ = 0.16 nA) (**d**) and $CO_2$ physisorption structure at 0.1 mbar $CO_2$ ($V_t$ = 0.54 V; $I_t$ = 0.16 nA) (**e**) on the Rh(111) surfaces. **f–h** Corresponding atomistic ball model illustrations of the observed NAP-STM images. Dark blue, black and red balls represent Rh, C and O atoms, respectively.

orders between adsorbate CO molecules and surface Rh atoms. So, the transferred charge distributions of directly bound t-CO at the atop site would have a higher probability of negative charge density than that of h-CO at the hollow site on the Rh(111) surface. This insignificant difference hinders characterizations of confined adsorption structures in direct observations[27]. Our observation results not only agree with previously reported dense CO/Rh(111) structures under a CO environment of approximately 1 bar, but also unravel a distinctive feature of the dominantly bright site-specific tunneling probability, depending on the perpendicular z-direction of distance between STM tip and Rh surface in measurement conditions (Supplementary Fig. 1).

Attempting the direct observation of molecular behaviors of $CO_2$ at the interface is more challenging than working with CO/ Rh, as displayed in Fig. 1e. This representative NAP-STM image under 0.1 mbar $CO_2$ conditions was taken after being filled with $CO_2$ gas for approximately 5 min, where the bright blobs, indicated by arrows, randomly diffused on the surface[34]. We note that our real-space probing of quickly moving $CO_2$ molecules on the Rh surface was not able to take clear topographic images exploiting the same optimized measurement conditions of CO/Rh(111). So, we adopted the STM tip movement in fast-scanning mode with a few hundred milli-seconds for the local area of taken under $CO_2$ environments at 300 K. Weakly bound $CO_2$ molecules on the Rh(111) surface did not show any ordering of intermolecular interactions with each other during observations for 1 h; the randomly aligned l-$CO_2$ molecules via the physisorption process are depicted as Fig. 1h (Supplementary Fig. 2). The trend of momentary diffusion of the physisorbed $CO_2$ molecules could be distinguished in consecutively recorded NAP-STM images with a time interval of 2.8 s (Supplementary Fig. 3), and the characterized immobile chemisorption $CO_2$ (ads.) molecules keep their ellipsoidal shape with a lateral size of 5.4 ± 0.4 Å and height of 0.3 ± 0.1 Å in real-space measurements. Moreover, the $CO_2$ (ads.) species that appear are absolutely distinct from the impurity (lateral size: 12 Å; height 1.0 Å) on the observed local area, which could be easily isolated of the visible difference along the z-axis in three-dimensional space according to a simplified Wentzel–Kramers–Brillouin (WKB) approximation[35]. In addition, the characterized $CO_2$ (ads.) molecules are consistently observed at different tunneling conditions in NAP-

STM measurements, showing that the recorded tunneling images of $CO_2$−Rh interface analysis results are far from a tip-induced artifact in direct observations (Supplementary Fig. 4).

This exposes a characteristic of $CO_2$ adsorption not found in CO molecules. In one aspect of molecular orbital configurations, the chemisorption of CO/Rh(111) is an exothermic process via π-backdonation in which electrons transfer from surface Rh atoms to CO molecules to relieve excess negative charge[33]. However, the ground state of $CO_2$ is similar to the linear dimer molecule with $^1\sum_g^+$ symmetry[6]. This inactive electronic configuration of the $CO_2$ leads exclusively to the physisorption process on the Rh surface, and the chemisorption of $CO_2$ needs prerequisite energy to overcome the potential barrier[36]. Therefore, the intra-valency-shell orbitals of $CO_2$ should be filled with induced-electrons at the moment of molecular collisions on the surface to proceed with the chemisorption process via modification as an active form of electronic configuration (Supplementary Fig. 5). The Walsh diagram[37] explains this relationship between orbital energy constructions and $CO_2$ molecular geometries of linear and bent structures; applicable predictions based on ab initio calculations imply the possibility of a metastable $CO_2^{\delta-}$ formation with a relatively long lifetime of 90 μs at molecular collisions[38].

**X-ray spectroscopy analysis of chemical binding energy.** Topographic analysis results provide intuitive knowledge of adsorbate-driven chemisorption structures at the interface. Still, the acquired local density of states images have technical limitations on interpreting the chemical information of adsorbate species at the same moment during the observations. To investigate the observed chemisorption species in real-space, further chemical binding energy analysis was also performed under environmental conditions using synchrotron-based NAP-XPS. The incident X-ray photon energy for each taken core-level spectrum was selected to 400 eV (Rh 3d and C 1s) and 640 eV (O 1s) in order to collect escaping photoelectrons at a probing depth of 5–10 Å from the topmost Rh layer[39].

Figure 2a exhibits acquired Rh 3d core-level spectra under UHV, 0.1 mbar CO and 0.1 mbar $CO_2$ environments. After repeated cycles of the flat Rh surface preparation (Supplementary Fig. 6), the obtained spectrum (labeled as clean) shows two

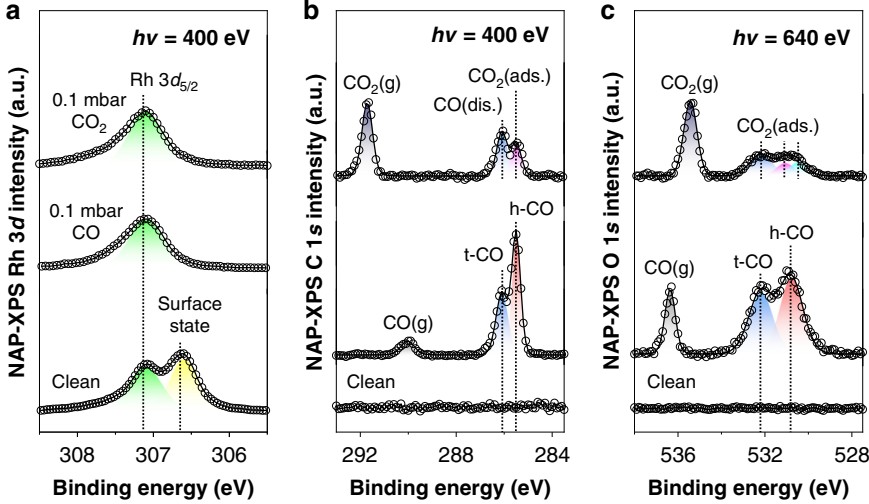

**Fig. 2 Chemical binding energy analysis for adsorbed CO and CO$_2$ molecules. a−c** Synchrotron-radiation NAP-XPS measurements for Rh 3*d* ($h\nu$ = 400 eV) (**a**), C 1*s* ($h\nu$ = 400 eV) (**b**) and O 1*s* ($h\nu$ = 640 eV) (**c**) core-level spectra under UHV, 0.1 mbar CO and 0.1 mbar CO$_2$ environments. The metallic Rh 3$d_{5/2}$ core-level and non-perturbed surface state of Rh atoms deconvoluted with green and yellow peaks, respectively. Site-specific molecular CO bound properties of t-CO (blue) and h-CO (red) indicated in both of C 1*s* and O 1*s* core-level spectra. Chemical binding features of dissociative adsorption for CO$_2$ at the interface also resolved individually with colors of blue, magenta and cyan, respectively.

distinctive peaks, attributed to a nonperturbate surface state and bulk Rh 3$d_{5/2}$ metal atoms at 306.6 eV (yellow color) and 307.1 eV (green color), respectively[40,41]. In gaseous environments, the intense surface state peak completely disappears, regardless of the different kinds of gas molecules in Rh 3*d* core-level spectra since the surrounding adsorbates of *s*- and *p*-orbitals have strongly correlated hybridization interactions nearby surface Rh atoms of *d*-band electronic structure[42]. This indicates that the adsorbate molecules modify the electronic structure of surface regime including first- and second-layer Rh atoms, resulting in surface core-level shift (SCLS) phenomenon[43]. Their delicate spectral changes, depending on the different kinds of adsorbate gas molecules (i.e. CO or CO$_2$) at NAP is analyzed in the comparison plots of Rh 3*d* core-level spectra taken in each different gaseous environment (Supplementary Fig. 7). The adsorption of gas molecules and their effective collision behaviors on the Rh(111) surface commonly make a noticeable broadening of the peak at 307.5 eV beside the characterized portion of Rh bulk species in the Rh 3*d* core-level spectra. In particular, we can find a small spectral shoulder at 307.9 eV, as displayed in the overlapping comparison plot of CO(g) and CO$_2$(g) in Supplementary Fig. 7, which implies that the surface Rh 3*d* core-level shifts get involved in the adsorbate–Rh atoms bonding formation properties by reactive molecule collisions and electronic charge redistribution on the Rh(111) surface[44–46].

Figure 2b, c exhibits spectroscopic discrepancies of adsorbate CO and CO$_2$ on the Rh(111) surface. Under 0.1 mbar CO conditions, we detected two kinds of adsorbate species and one gaseous species at each different peak position in deconvoluted C 1*s* and O 1*s* core-level spectra. This metal carbonyl formation (M-CO) could be examined by individual metal element electron transfers binding the light organic molecule and the metal atom by the $\pi$-backdonation synergic process[47]. Spectral evolving features ascribe the chemical bonding of CO/Rh for hollow sites (h-CO; red color), atop sites (t-CO; blue color), and free molecules CO(g) (light gray color). Those assigned adsorbate CO species of peaks at 285.5 and 286.1 eV for C 1*s* core-level and at 530.8 and 532.2 eV for O 1*s* core-level spectra[40,48] have a point of similarity in the greater signal intensity of h-CO rather than t-CO, where measured chemisorption properties are consistent with the literature for CO/Rh(111) study at a maximum $\theta_{CO}$ of

0.75 ML. Moreover, the relative C 1*s* signal intensity ratio (Supplementary Table 1) of t-CO/h-CO is 0.51, and this spectroscopic evidence supports the topographic observation result of (2×2)-3CO structure in Fig. 1d. However, the characterized ratio of t-CO/h-CO in the C 1*s* core-level spectrum is not matched with the calculated ratio in the O 1*s* core-level spectrum (0.80). Because the collected signal intensities would be influenced by the kinetic energy of irradiated photon beams during the capturing of photo-emitted electrons at the interface. It does not mean the qualitative change of characterized adsorbate species during the NAP-XPS measurements. The populated photoelectrons could be measured with different signal intensity ratios of t-CO/h-CO between C 1*s* and O 1*s* core-level spectra by the influence of adsorbate geometry and photoelectron diffraction effect as a function of photon energy, as reported previous literature[40,48,49].

The well-characterized spectroscopic trends for CO−Rh interactions do not match the obtained C 1*s* and O 1*s* core-level spectra under 0.1 mbar CO$_2$ environments. Interestingly, two peaks of the assigned CO$_2$ (ads.) and CO (dis.) were identified at 285.5 and 286.1 eV. They are corresponding to chemisorbed CO$_2$ and dissociated CO adsorbates in C 1*s* core-level spectrum, which have the same deconvolution profiles as at full width at half-maximum (FWHM) and peak center positions at the binding energy for the analysis results of CO/Rh(111). However, the effective collisions of CO$_2$ free molecules on the Rh (111) surface would form weakly bound CO$_2$ (ads.) species[20,22], as observed in Fig. 1e, which differs remarkably from the measured NAP-XPS results in the CO(g) environment. This unique property is characterized at the early stage of gaseous CO$_2$ exposure, the assigned peak of CO$_2^{\delta-}$ at 289.4 eV in C 1*s* core-level spectrum (Supplementary Fig. 8) suggests a clear evidence of molecular CO$_2$ interactions with Rh atoms at NAP. The signal intensity ratio of t-CO/h-CO goes in the opposite direction from the spectral acquisition under CO(g) conditions. We note that the peak intensity ratio of gas phase/adsorbates is critically influenced by sample-to-aperture distance, which is irrelevant to the qualitative characterization of adsorbate species in NAP-XP spectra under CO(g) and CO$_2$(g) environments. Furthermore, we could not obtain the unsuspected tendency in C 1*s* and O 1*s* core-level spectra until approximately 100 min of CO$_2$ gas

exposure, indicating that the chemical shifts of C 1s and O 1s start to be visible after long exposure to $CO_2$.

At this point, an additional peak appears at 530.5 eV (O1) in O 1s core-level spectrum, besides the $CO_2$ (ads.) species for hollow (O2) and atop (O3) sites of the Rh surface (Supplementary Fig. 9). A noticeable O1 peak could also be associated with the intramolecular chemical bonding of *COO−Rh, which makes a segmented dissociative adsorption neither t-CO nor h-CO of transient b-$CO_2$ at the Rh interface. We can find a clear spectral broadening feature between initial ($t_0 + 14$ min) and equilibrium ($t_0 + 132$ min) O 1s core-level spectra in time-lapse NAP-XPS measurements, and the deconvoluted O1 peak is distinct from an adjacent O2 species with binding energy differences of 0.5−0.6 eV (Supplementary Fig. 10). Moreover, the evolved intermediate species cannot be classified as an atomic O* at the hollow site of Rh(111) surface in the literature[50,51]. Taken together, the unusual features under $CO_2$(g) conditions imply that the partial chemical species of adsorbate molecules may occupy atop, bridge, and hollow sites on the Rh surface as intermediates. Consequently, the intramolecular bond of O−CO* loosens by pulling both sides of the chemical bond with surface Rh atoms; in other words, electrophilic surface Rh atoms strongly attract the lone-pair electrons of O−CO*, and the other side of molecular bonding for b-$CO_2$ would consolidate a binding of CO* preferentially at the atop site on the Rh surface. At equilibrium, it is possible for dissociated O*, CO* and chemisorbed b-$CO_2$ to exist together at the same Rh terrace; meanwhile, the resolved peaks of $CO_2$ (ads.) species at a binding energy of 530.5 (cyan color) and 531.1 eV (magenta color) in O 1s core-level spectra have peak shifts of +0.3 and +0.1 eV as time lapsed in NAP-XPS measurements (Supplementary Table 2). Nonetheless, these specific observations only elucidate the influence of partial chemical binding of *O −Rh at the interface, because the direct molecular bonding information of *$CO_2$−Rh could emerge from charge transfers for the intramolecular property of *CO−Rh species during the $CO_2$ chemisorption process.

As a result, the measured peak intensity ratio (C2/C1) in C 1s core-level spectra under 0.1 mbar $CO_2$ environment increases 172.9% between initiation and equilibrium (Supplementary Table 1), because the spectral portion of dissociated CO contributes to the change of relative signal intensity ratio. The corresponding relative peak intensity ratio of O3/O2 in O 1s core-level spectra of initial and equilibrium also increased 160.0%, indicating that the chemical species interpretation using a widely used peak deconvolution procedure for NAP-XP spectra are thereby reliable as supporting evidence of the $CO_2$ dissociation process. We emphasize that the represented spectroscopic evidence of dissociated CO* from $CO_2$ (ads.) was obtained in the strictly managed X-ray photoemission experimental setup to exclude the issue of photon-induced contaminations[52]. The high-flux X-ray photon beam was not continuously irradiated to the Rh(111) single crystal proportional to the exposure time of $CO_2$ gas molecules in the analysis chamber. No significant evolution of carbon fragment or carbidic species was found in the C 1s core-level spectra at NAP conditions, which is also confirmed in the plotted comparison spectra of before and after pump down (Supplementary Fig. 11).

**Direct observations of $CO_2$ dissociation on the Rh surface.** Both measurements of electron microscopy and X-ray spectroscopy techniques commonly experienced unexpectedly long $CO_2$ gas exposure to reach equilibrium, in contrast with the immediately attained equilibrium conditions in a CO environment. We emphasize that these confusing results are neither contamination of CO impurity in $CO_2$(g) nor partial thermolysis of $CO_2$ molecules by glowing tungsten filaments from the analytical instruments; such a hypothesis for slowly conducting CO contamination on Rh surfaces cannot clearly explain the opposite trend of t-CO/h-CO ratio for C 1s core-level and the existence of O (ads.) species for O 1s core-level NAP-XPS spectra in a $CO_2$ environment. To discover the detailed interactions at the $CO_2$−Rh interface, time-lapse in situ observations were carried out during the $CO_2$ dissociation over Rh(111) surface.

Figure 3a shows an NAP-STM image composed of bright and dark spots at $CO_2$(g) equilibrium, where a marked parallelogram shape connects to each corner of the periodic bright spots, and the arrows indicate randomly distributed dark spots. The

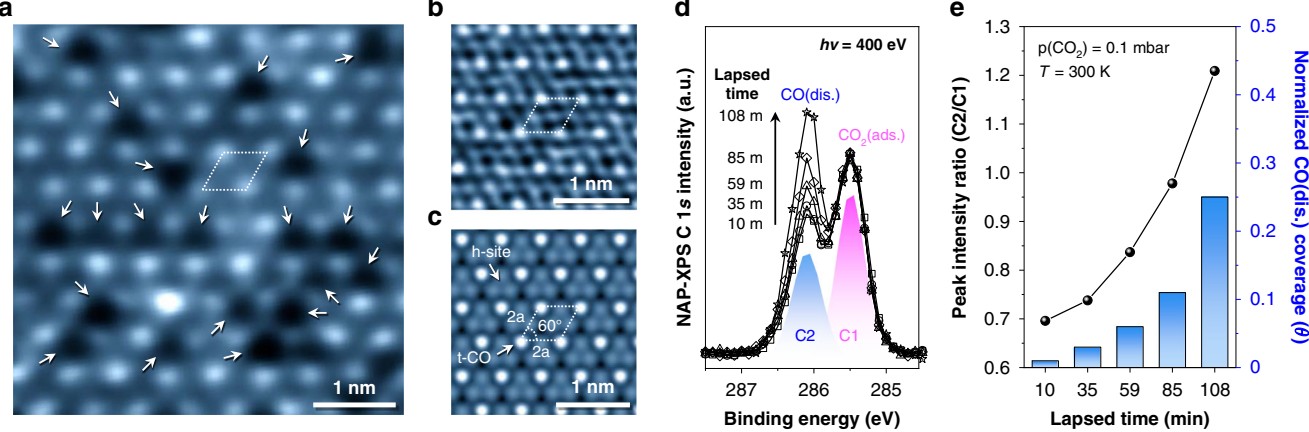

**Fig. 3 Direct observations of dissociated CO and O adsorptions at the $CO_2$/Rh interface. a−c** Atom-resolved adsorption structure analysis during the $CO_2$ dissociation over Rh(111) surface. A representative real-time observation result using NAP-STM at 0.1 mbar $CO_2$ ($V_t = 0.25$ V; $I_t = 0.17$ nA) (**a**), their narrow-scale image ($V_t = 0.21$ V; $I_t = 0.30$ nA) (**b**) and the DFT-simulated STM image of (2 × 2)-CO adsorption structure on the Rh(111) surface (**c**). Indicated arrows and corners of the parallelogram on the images are corresponding to dissociated O* and CO*, respectively. **d, e** Time-lapsed spectroscopic measurement results of dissociated CO adsorptions on the Rh(111) surface using NAP-XPS ($h\nu = 400$ eV). Recorded C 1s core-level spectra on the Rh(111) surface (**d**) and peak intensity ratio histogram analysis plots for the measured C 1s spectra (**e**) under 0.1 mbar $CO_2$ environment. Each peak intensity of C2 was normalized to the of fixed C1 ratio in C 1s core-level spectrum. The assigned CO (dis.) and $CO_2$ (ads.) peaks are labeled as C2 (blue) and C1 (magenta), respectively. All spectral interpretation procedures were identically carried out before and after normalization of C 1s core-level NAP-XP spectra (Supplementary Fig. 13).

measured nearest-neighborhood distance of bright spots is approximately 5.4 Å, which is equivalent to double the atomic length of Rh−Rh on the surface[27,53]. Surprisingly, indicated bright and dark spots exactly occupy the atop and hollow for each site of the Rh surface at the measured local area; a substrate Rh atom-resolved narrow-scale image, as displayed in Fig. 3b, demonstrates that the indication of the parallelogram correlates with the dotted connections of each corner for a bright spot in Fig. 3a. Considering the NAP-STM images, we suggest the reasonable assignments of bright and dark spots corresponding to CO* at the atop site and O* at the hollow site of surface Rh atoms, which is critical evidence of spontaneous $CO_2$ dissociation over the Rh surface at equilibrium. In fact, the ordering structure formation of $(2 \times 2)$-CO in Fig. 3b matched remarkably with a visualized DFT-based STM simulation image for $(2 \times 2)$-CO/Rh (111) in Fig. 3c; in addition, the dissociated O* was typically observed as dark spots, owing to the electron depression effect during the empty-state of surface observations using the tungsten STM tip[54,55]. On the other hand, the observed dark spots are irregularly scattered on the Rh(111) surface, which implies that some dissociated atomic oxygen species may be effectively removed rather than the ordered *CO−Rh by nearby adsorbates via the thermodynamically preferred desorption process of $O_2$(g) or $CO_2$(g). We note that these clearly distinguished CO* and O* species on Fig. 3a, b happen to be observed after the $CO_2$ dissociation process on the Rh(111) surface. Once the catalytic reaction initiates, the $CO_2$ (ads.) molecules have complex interactions with surface Rh atoms, represented by randomly tangled structures by chemisorbed, dissociated, and intermediate forms of $CO_2$ (ads.) at the same time (Supplementary Fig. 12).

As well as time-lapse NAP-XPS measurements for C 1s, core-level spectra in a $CO_2$ environment (Fig. 3d) display the relative growth of intensity for CO (dis.) species (C2) compared to the other assigned peak for $CO_2$ (ads.) (C1) in deconvolution areas. The sequentially recorded spectra in order of time show that the only CO(dis.) peak increases proportionally to the elapsed time in spectra. The plot of the peak intensity ratio of C2/C1 (black color) and histogram analysis (blue color) of the normalized CO(dis.) coverage (the plot of unnormalized C 1s core-level spectra is provided in Supplementary Fig. 13) exhibits the number of relative variations between dissociated CO* and $CO_2$ (ads.) on the Rh surface at 300 K under 0.1 mbar $CO_2$(g) as displayed in Fig. 3e. Given the plot, the measured coverage of dissociated CO* from $CO_2$ (ads.) is approximately 0.1 ML after introducing gaseous $CO_2$ molecules into the analysis chamber 85 min later. In consequence, the catalytic $CO_2$ dissociation does not reach equilibrium until 108 min in our direct observation results of 0.25 ML at maximum CO coverage. This unexpected phenomenon is feasible for the molecular $CO_2$ conversion occurring over the Rh catalyst, but the addressed detectable measurements strongly support the exothermic bond-breaking process for the $CO_2$ dissociation after all.

In principle, reaching CO (dis.) coverage of 0.1 ML would require a few million years[17] because of the low probability of dissociative adsorption at the $CO_2$(g) pressure in the $10^{-6}$ mbar range (Supplementary Fig. 14). On the other hand, the reaction requisition time, which varies between a few hundreds of micro-seconds and one month, could be dramatically reduced, depending on the reaction temperature at 1 bar of $CO_2$(g). Discrepancies between fundamental theory and empirical calculations can occur as a result of underestimating experimental kinetic data. Nevertheless, an enormous turnover time gap during the direct observations of $CO_2$ (ads.) at 0.1 mbar $CO_2$ would be engaged in increased effective molecular collisions at $CO_2$−Rh interface[14,16]. The turnover frequency of chemisorbed $CO_2$ molecules is determined by the combined equations of a Maxwell−Boltzmann function at a certain temperature and probabilities for dissociative adsorption of $CO_2$; as a result, the increased chemical potential at the elevated pressure facilitates the dissociative chemisorption state beyond the potential energy barrier[19,56]. Therefore, in industrial chemical reactors for $CO_2$ conversion typically operating at high pressure, dissociated CO* molecules are effectively involved in the catalytic reaction process at the early stage of $CO_2$RR at the Rh catalyst interface, whereas the intermediate product of formate is selectively hindered at the projected reaction pathway.

**Electronic charge analysis and proposed mechanism.** The experimental results of atom-resolved microscopy and X-ray spectroscopy indicate that chemisorbed b-$CO_2$ molecules spontaneously break up at the $CO_2$−Rh interface, which would lead to selective yields of intermediate species developing further complicated catalytic reaction processes. DFT calculations were performed to elucidate the details of electronic charge distribution between the b-$CO_2$ molecule and surface Rh atoms by the Bader charge analysis method based on a theory of atoms in molecules (AIM)[57], and to predict a catalytic reaction route of $CO_2$ dissociation on the Rh(111) surface. The calculated molecular geometry for $CO_2$ in the gas phase has a bond angle ($\angle O = C = O$) of $178-180°$ and an intramolecular distance ($d_{O-C}$) of $1.172-1.181$ Å, whereas the activated $CO_2^{\delta-}$ molecules at the interface could have transformed structures, as reported in the literature[20,58,59].

In Fig. 4a, b, the optimized molecular geometry at local minimum ($\Delta E_{ads} = -0.39$ eV) has a bond angle of 133.3° and coupled distances of $d_{Rh-C}$ and $d_{Rh-O}$ corresponding to 2.059 and 2.145 Å, respectively. The surface Rh atoms bind with adjacent intramolecular C and O atoms of b-$CO_2$. Meanwhile, the projected nonuniform charge distributions would induce separately to the gradient points of positively charged (Rh$_1$−C and Rh$_2$−O) or negatively charged (O−CO) atoms. Finally, a net charge of $-0.51$ e is transferred from the Rh atoms to the b-$CO_2$ molecule. The electron density iso-surface at a density of 0.004 e/Å$^3$ confirms the chemical interaction formed between b-$CO_2$ and the Rh surface, which initiates the subsequent spontaneous $CO_2$ dissociation process by O−CO bond cleavage at the interface. Figure 4c presents the energetics of overall catalytic $CO_2$ dissociation over the Rh(111) surface, which is exothermic ($\Delta E_{diss.} = -0.82$ eV) with an activation energy barrier, $\Delta E_{TS}$ of $+0.58$ eV. As a result, the dissociated CO* and O* remain on the Rh surface (Supplementary Fig. 15). The calculated final dissociated state exhibits $7-11\%$ shorter Rh−C (1.834 Å) and Rh−O (1.987 Å) bond distances while the moderate level of $\Delta E_{TS}$ indicates the occurrence of complicated atomic rearrangements at the transition state. That is, a partial *CO intramolecular bonding of Rh−COO* at atop Rh$_2$ site would be rearranged to the bridge Rh$_2$ site. Eventually, the dissociated O* is relocated to the most thermodynamically preferred hollow site by the bond cleavage of *O−CO. This proposed chemical reaction route confirms that the strongly correlated electronic interactions between chemisorbed $CO_2$ molecules and Rh(111) surface possibly lead to the thermodynamic driving force of $CO_2$ dissociation at the Rh interface in realistic reaction conditions. The DFT-calculated average binding energy of four $CO_2$ molecules adsorbed on Rh (111) was decreased to $-0.19$ eV/$CO_2$ (Supplementary Fig. 16). The multiple $CO_2$ binding configuration with the weaker binding energy would have a limited survival under the higher $CO_2$ partial pressure, providing the increased $CO_2$ surface coverage. We note that the DFT-estimated $\Delta E_{TS}$ values of $CO_2$ dissociation on Rh (111) with conventional generalized gradient approximation level exchange-correlation functionals are usually underestimated from the experimental value (Supplementary Table 3)[59−62]. However,

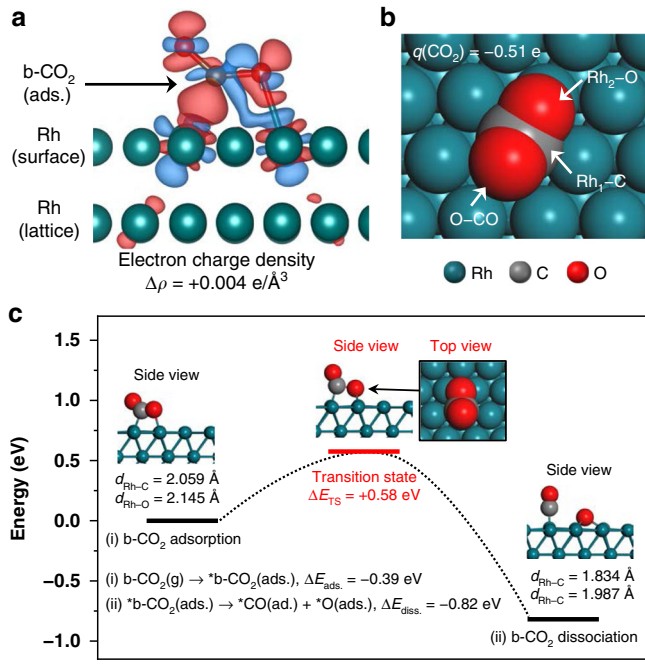

**Fig. 4 Electronic interaction analysis and proposed mechanism for CO₂ dissociation. a** Electron iso-surface ($\rho = 0.004$ e/Å³) presenting the distribution and morphology of the orbitals participating in the chemical interaction between b-CO₂ (ads.) and Rh atoms, which visualizes that each positively charged adjacent surface Rh atom is binding with C (Rh₁−C; +0.03 e) or O (Rh₂−O; +0.17 e) atom of the b-CO₂ (ads.). Integrated electron charge density difference defined as $\Delta\rho = \rho_{total} - \rho_{surf} - \rho_{adsorbate}$ in Bader volume, where $\rho_{total}$, $\rho_{surf}$, and $\rho_{adsorbate}$ are charge density for b-CO₂/Rh, Rh surface, and b-CO₂ molecule, respectively. Charge accumulated or depleted areas in the iso-surface are presented as red or blue. **b** A Bader charge analysis result of b-CO₂/Rh. A net charge (q) of −0.51 e was transferred to b-CO₂ (ads.) from adjacent Rh surface atoms. **c** DFT-calculated pathway and energetics of CO₂ dissociation on the Rh(111) surface. (Rh, emerald; C, gray; O, red). A single imaginary frequency of 434i cm⁻¹ was calculated for the TS.

such underestimation does not significantly interfere with the theoretical analysis of the CO₂ dissociation mechanism.

## Discussion

We have revealed a chemical bond cleavage of *O−CO at CO₂−Rh(111) interface in reaction environments using NAP-STM and NAP-XPS techniques. Those direct observations of CO₂ dissociation, an essential step in primary energy source production processes, indicate that the active charge transfers between CO₂ molecules and metal surface at elevated pressures significantly affect yields of intermediate products at the early stage of catalytic reactions. Our combined experimental results clearly show ordered (2 × 2)-CO structure formation by spontaneous CO₂ dissociation at the Rh interface. Similarly, probed time-lapse evolutions of the dissociated CO* chemical binding at atop sites of the Rh surface support evidence of the observed catalytic reaction extensively along the same lines. For the constructed model of b-CO₂/Rh, which may be facilitated by the increased chemical potential of the Rh interface under realistic reaction conditions, the DFT calculations explain a possible chemical reaction route by activated charge transfer at the nonequilibrium state of b-CO₂ molecules on the Rh surface. Furthermore, the proposed CO₂ dissociation pathway not only suggests a thermodynamic preference of Rh−carbonyl formation toward

methane synthesis, but also describes a distinct reaction direction against yields of intermediate formate carrying on a route for methanol synthesis using Cu catalysts. Our combined investigations provide fundamental insights in determining the selectivity of energy source production at the molecular level and therein contributes to the rational design of catalytic factors for improving CO₂ utilization in broad applications of cutting-edge science in heterogeneous catalysis.

## Methods

**Sample preparation.** A commercially available single crystal Rh(111) sample (99.99%) was purchased from Mateck GmbH. The one-side polished (roughness < 10 nm) metal crystal was prepared by a floating zone method with a high cut-orientation accuracy < 0.1°. The prepared sample was cleaned by repeated cycles of Ar⁺ ion-bombardment sputtering ($P_{Ar} = 1 \times 10^{-5}$ mbar at 1.0 keV) and annealing to 1000−1050 K for 5 min in UHV. Freshly prepared contaminant-free surface characterization was confirmed using STM, low-electron energy diffraction (LEED), and XPS.

**NAP-STM experiments.** Topographic NAP-STM images were recorded in a small volume (15 mL) reaction cell-integrated STM scanner (Aarhus STM 150 NAP, SPECS GmbH), where the enclosed volume could be physically separated with a UHV chamber (base pressure: $1 \times 10^{-10}$ mbar) by two locking screws[63]. The cleaned sample was transferred to the pressure and temperature variable reaction cell, taking direct observation results under gaseous environments. High-purity (99.999%) CO or CO₂ gas molecules were further purified to remove infinitesimal amounts of Ni or Fe-carbonyl and water using gas purification filters (PALL Corp.) in the multi-gas delivery manifold system, which was introduced to the reaction cell by a precision leakage valve. A chemically etched tungsten tip was employed to record tunneling current between the sample and the STM tip. Sharpening and cleaning processes of the tip were treated with the positively charged in situ noble gas ion sputtering method before STM measurements under the gas environment. The positive bias voltage was applied to the sample to prevent gas molecule-induced tip crash during the measurements at 300 K in constant current mode. Each empty-state topographic image of tunneling conditions was denoted as $V_t$ (applied voltage) and $I_t$ (tunneling current), respectively.

**Synchrotron-based NAP-XPS experiments.** All X-ray photoemission spectra were taken at the endstation of the BL-13B soft X-ray undulator beamline facility of the Photon Factory (PF) at the High Energy Accelerator Research Organization (KEK) in Japan[41]. Clean Rh surface preparation and photoemission core-level spectrum measurements were performed in each separate preparation (base pressure: $2 \times 10^{-10}$ mbar) and analysis (base pressure: $2 \times 10^{-9}$ mbar) chamber, respectively. Sample cleanliness and long-range ordering of Rh atoms on the surface were confirmed by a LEED instrument in UHV after the repeated cycles of Ar⁺ ion-bombardment sputtering and high-temperature annealing. In the same manner, no contaminants such as C, O, B, Ni and Si were detectable on the freshly cleaned Rh(111) surface by XPS. High-purity CO (99.999%) and CO₂ (99.999%) gas cylinders were connected to a compact–sized gas manifold. All gas lines had a bake out procedure at 383 K with a high-speed pump out for at least 12 h before the cleaned gas feed in experiments. CO or CO₂ gas molecules were backfilled with the analysis chamber using a precision leakage valve, which introduced reactive gas molecules that were monitored with a quadrupole mass spectrometer (HAL-201, HIDEN) in the chamber of the first differential-pumping stage. A modified hemispherical electron energy analyzer (EA125HP, Omicron) was used for the NAP-XPS measurements. The selected incident photon energies used for NAP-XPS analysis were 400 eV (C 1s; Rh 3d) and 640 eV (O 1s), to take almost the same surface sensitivity of each core-level spectrum at KE of approximately 100 eV (probed depth from the topmost layer ≤10 Å). The high-flux photon beam was irradiated to the Rh(111) model catalyst within 3 min at a shot of the selected core-level analysis, then the beam shutter was closed immediately after the acquisition of each spectrum. X-ray beam-induced sample damage and influence of photo-ionization effect were not detected during the NAP-XPS measurements. The acquired spectra of binding energy were calibrated by the Fermi-edge of the sample substrate at the incident photon energy. For detailed analysis of acquired core-level XP spectra, each spectrum was subtracted by a Shirley-type background that measured peaks fitting a widely accepted mixed ratio (70%:30%) of Gaussian −Lorentzian function using the CasaXPS package.

**DFT calculations.** All spin-polarized DFT calculations for the CO or CO₂/Rh systems were performed using the Vienna ab initio simulation package (VASP), and the interaction between the valence electrons and the ionic core was described by the projector augmented wave (PAW) method[64,65]. The exchange-correlation energy of the Kohn−Sham equation was functionalized with the revised Perdew−Burke−Ernzerhof (RPBE) functional[66]. A 6 × 6 × 4 supercell was used to describe the Rh(111) surface (Supplementary Fig. 17). The calculated lattice parameter of Rh was 3.85 Å (refer to Supplementary Table 4 for more discussions).

The bottom two Rh layers were fixed upon geometry optimization. A $4 \times 4 \times 4$ $\Gamma$-centered $k$-point grid was used to sample the first Brillouin zone. The mechanism of the $CO_2$ dissociation process over an Rh surface was studied using a model structure of single $CO_2$ molecule/Rh(111). Appropriate multiple molecular binding configurations were also estimated to reproduce the NAP-STM images. Valence electron functions were extended with the plane-wave basis to an energy cutoff of 400 eV, and a 15 Å vacuum space along the $z$-direction was applied to avoid interactions between the repeating slabs. The convergence criteria for the electronic structure and the geometry were set to $10^{-3}$ eV and 0.05 eV/Å, respectively. The van der Waals (vdW)-corrected DFT-D3 method with the Becke−Johnson damping model was applied for all calculations[67]. Gaussian smearing function with a finite temperature width of 0.02 eV was used. The location and energy of transition state (TS) were calculated with the climbing-image nudged elastic band method[68,69]. Subsequent normal mode analysis was performed to confirm the TS. Simulated STM images were acquired by integrating the occupied states at and below the Fermi-level (down to $E_F - E \leq 0.5$ eV). The calculated adsorption energy ($E_{ads}$) is defined as $E_{ads} = E_{total} - E_{surf} - E_{adsorbate}$, where $E_{total}$ is the total energy for the model system with adsorbate species; $E_{surf}$ is the total energy for the optimized bare surface; and $E_{adsorbate}$ is the total energy for the adsorbate species.

## Data availability

The data that support the findings of this study are available from the corresponding authors upon reasonable request.

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

## Acknowledgements

This work was supported by the Institute for Basic Science (IBS) [IBS-R004]. H.H., H.Y.K., and B.S.M. are thankful for financial support from the National Research Foundation of Korea (NRF), funded by the Ministry of Science and ICT (NRF-2019R1A2C1089256, NRF-2019R1A2C2008052, NRF-2015R1A5A1009962). B.S.M. would like to thank the GRI project of GIST 2020 for their support. This research used resources from the Center for Functional Nanomaterials (CFN), which is a U.S. DOE Office of Science Facility, and the Scientific Data and Computing Center, a component of the Computational Science Initiative, at Brookhaven National Laboratory (BNL) under Contract No. DE-SC0012704. The synchrotron-based NAP-XPS experiments were performed under the approval of the Photon Factory Program Advisory Committee (PF PAC No. 2018S2-005).

## Author contributions

J.K. and J.Y.P. conceived and designed experiments. J.K. performed NAP-STM measurements. W.H.D. and K.U. performed LEED and NAP-XPS measurements. K.M., H.K. and B.S.M. supervised the NAP-XPS measurements at the synchrotron beamline facility. H.H. and H.Y.K. performed theoretical calculations. J.Y.P., H.Y.K., and B.S.M. supervised the project. J.K., H.H., and W.H.D. wrote the paper. All authors discussed the experimental and theoretical calculation results for manuscript preparation.

## Competing interests

The authors declare no competing interests.
