## [Peer Review File · Nature Communications]

REVIEWER COMMENTS

Reviewer #1 (Remarks to the Author):

The present work entitled „How Rh surface breaks CO₂ molecules under ambient pressure” by Park and coworkers discusses cutting edge results of current catalysis research using near-ambient pressure scanning tunneling microscopy and x-ray photoelectron spectroscopy complemented by density-functional theory (DFT) calculations. The manuscript is of high quality, the figures excellent and the organization of the manuscript is excellent. Nevertheless, some points should be addressed, especially with respect to DFT calculations.

- 1) The RPBE (revised Perdew-Burke-Ernzerhof) exchange-correlation functional is known to overestimate lattice constants drastically; see Stroppa, A.; Kresse, G. *New J. Phys.* 2008, 10, 063020. Which lattice parameters have been adopted for the Rh surface? Authors, please clarify and consider that lattice parameters off equilibrium would eventually lead to a spurious activation of the surface (model).
- 2) With respect to CO₂ dissociation, I consider the work by Abbott and Harrison as important and related to the present work. The authors should cite *J. Phys. Chem. C* 2007, 111, 13137-13148 and put results into context of the previous work by Abbott and Harrison. Although, the RPBE functional was found by other workers to perform rather ok for activation barriers it still features an underestimation of barrier heights of several kcal/mol (depending on the species "transferred in the reaction") typical of semilocal (GGA-type) functionals (see e.g. Truhlar and coworkers; *J. Chem. Phys.* 132, 164117 (2010)).
- 3) The computed reaction pathway of course shows a structure corresponding to the transition state. It is required to do a normal mode analysis in order to detect the “reaction coordinate” featuring only one imaginary frequency. The authors are advised to report this mode and frequency.
- 4) The manuscript discusses of course results addressing the pressure gap of classic surface science. The calculations seem to be for a more dilute situation. The authors are advised to report the cell size or the coverage of the simulation cell for which the reaction pathway was calculated. The authors are also advised to comment on incorporating the effect of the elevated pressure in their simulations.

I recommend publication once these comments have been addressed.

Sincerely,
Joachim Paier

Reviewer #2 (Remarks to the Author):

In the manuscript titled “How Rh surface breaks CO₂ molecules under ambient pressure”, the authors studied the activation of CO₂ on Rh(111) with XPS and STM under near-ambient pressure(NAP) and theoretical calculation. They evidenced the dissociation of CO₂ to form CO and O species on the Rh surface at room temperature. The observation of CO₂ dissociation with NAP techniques was studied and

reported in the past several years (PNAS, 2017, 114, 266706–6711; J. Am. Chem. Soc. 2016, 138, 13246–13252; J. Am. Chem. Soc. 2016, 138, 4146–4154). For example, Eren et al detected dissociated species with NAP-XPS (J. Am. Chem. Soc. 2016, 138, 8207–8211) and Kim et al proved CO₂ dissociation using infrared reflection absorption spectroscopy (ACS Catal. 2016, 6, 1037–1044). This work does not provide much new insights on CO₂ activation although the authors obtained more information with theoretical calculation. I do not recommend its publication in Nature Communications. Some specific comments are listed in the following:

1. The above mentioned previous studies represent the recent progress in CO₂ activation study and are closely related to this work. The authors may need to cite them in the manuscript.
2. In the abstract, the authors claimed that “Here, we report direct observations of spontaneous CO₂ dissociation over the model rhodium (Rh) catalyst in realistic environments.” Normally, “realistic environments” represents the reaction conditions employed in industry, for example 50-100 bar for CO₂ hydrogenation. There still exists a pressure gap between NAP and the realistic reaction condition. Therefore, it is inappropriate to use the phrase of “realistic environments” here.
3. In page 3, the authors concluded that “Here, we report direct observations of chemisorption and dissociation processes at CO₂-Rh(111) interface.” However, the authors did not provide any experimental evidences for the intermediates during the dissociation processes. The observed CO and O species can only prove the occurrence of CO₂ dissociation. In this regard, the authors did not directly observe the dissociation processes.
4. In the NAP-STM results, the authors attributed the bright blobs in Fig. 1e to the physisorbed linear CO₂ molecules. In fact, under the condition of 0.1 mbar CO₂, the bright blobs can be possibly caused by other impurities. The corresponding NAP-XPS spectra should be provided to confirm the NAP-STM assignment.
5. In the NAP-XPS results, Fig. 2a shows the pressure of 0.1 Torr whereas in the manuscript, it was described that the experiments were carried out at 0.1 mbar CO₂ and 0.1 mbar CO. Besides, the Rh core level spectra in CO₂ and CO environments only showed the fitted green peak. However, it can be clearly found that there exist other intensities at the left side of the green peak. The author may want to provide some discussions.
6. In Fig 2b and 2c, both the C1s and O1s spectra show the chemisorption of t-CO and h-CO at 0.1 mbar. In principle, the signal intensity ratio of t-CO/h-CO in C1s should be the same with that obtained from O1s with the same gas exposure. As indicated by the authors, the relative C 1s signal intensity ratio of t-CO/h-CO is about 0.5, while it is close to 1 from O1s results. The authors may provide the calculated t-CO/h-CO ratio for O1s and give the explanation?
7. To elucidate the formation of O1 in Fig. 2c, the authors sequentially acquired O 1s core-level with different gas exposure (Supplementary Figure 5). They reported the peak shifts of O1 and O2 from their fitting results. However, the peak positions of O1 and O2 are very close to each other. Thus, the fitted results for peak shift could be artificial. It would be more reliable if the authors refer to the corresponding peaks ratios in C1s for O1s peaks analysis.

In Fig. 4d, the peak of C1 was assigned to chemisorbed CO₂, but the STM results in Fig. 4a and b do not show CO₂ chemisorption. Although the authors observed the ordered structure in Supplementary Figure 2, it could be caused by the variation of STM tip state. The authors may need to provide other evidence for chemisorbed CO₂.

Reviewer #3 (Remarks to the Author):

This paper by Kim et al. presents a novel study of the interaction of moderate pressures of CO₂ with an active Rh surface using advanced near-ambient pressure microscopy and spectroscopy. The subject matter is of keen interest both to the traditional surface science community and also to a wider audience with its clear relevance to catalysis.

There are, however, a few matters that need to be addressed in my opinion before this manuscript is ready for publication, these are detailed below:

- The issue of photon-induced reactions/cracking is raised by the authors, but is not clearly ruled out in their data, which could have consequences for the conclusions they draw. High flux density soft X-rays, especially those generated by modern insertion device beamlines are known to routinely lead to cracking of background CO inside UHV chambers leading to carbon build-up on the sample under investigation, along with other deleterious changes. Such beam-induced modifications are even more apparent in ambient pressure XPS, and I think the authors need to demonstrate thoroughly through a suitable control study that such effects are not the cause of the changes they are seeing. One relatively simple study would be to repeat the high pressure CO₂ long exposure experiment in the absence of the photon beam to ensure that the same result is gained.
- The gas feed purity is also essential in ambient pressure measurements as even trace impurities can cause misleading results, please could the authors explain in detail the steps they took to ensure that the gas feed was clean.
- Overall the assignments of the C1s XPS are not clearly explained, in my opinion – what is the identity of the so-called CO₂(dis) species in Figure 3? Is this distinct from what you are labelling *CO?
- It is claimed from the STM data in Figure 1 that there is physisorbed CO₂ (intact) adsorbed on the surface, but from the XPS data in Figure 2, only dissociated CO₂ is assigned, how do the authors reconcile this?
- What is the origin of the significant difference in gas phase peak heights in Figure 2? The CO peak is much lower in the C1s compared to the CO₂, however in the O 1s they are similar intensities, and the nominal gas pressure is the same.
- Tip-induced changes have been observed in high-pressure STM experiments before, do the authors have any comments regarding their likelihood in these experiments, and does the tip bias play a role in this?

- In figure 3, the C1s data has been normalised to the C1 peak, what is the justification for this? It would be good to present the unnormalized data in the supporting information to see the evolution of the total carbon species on the surface with time.

Overall this work underlines the challenges of the near ambient pressure techniques and the difficulties in attempting to correlate spectroscopic trends with atomically resolved microscopy. It would significantly benefit from some additional complementary spectroscopic probes such as RAIRS to aid in the identification of the C-O intermediate species.

Response to Reviewer #1:

General comment: *The present work entitled “How Rh surface breaks CO₂ molecules under ambient pressure” by Park and coworkers discusses cutting edge results of current catalysis research using near-ambient pressure scanning tunneling microscopy and x-ray photoelectron spectroscopy complemented by density-functional theory (DFT) calculations. The manuscript is of high quality, the figures excellent and the organization of the manuscript is excellent. Nevertheless, some points should be addressed, especially with respect to DFT calculations.*

Response: We appreciate the reviewer’s favorable evaluation of our research. We are glad to hear that the reviewer thinks that our manuscript is of high quality, and the figures and organization of the manuscript excellent. We have addressed some concerning points mentioned by the reviewer about presenting our DFT calculations in the revised manuscript, with point-by-point responses below.

Reviewer comment #1-1: *The RPBE (revised Perdew-Burke-Ernzerhof) exchange-correlation functional is known to overestimate lattice constants drastically; see Stroppa, A.; Kresse, G. *New J. Phys.* 2008, 10, 063020. Which lattice parameters have been adopted for the Rh surface? Authors, please clarify and consider that lattice parameters off equilibrium would eventually lead to a spurious activation of the surface (model).*

Response #1-1: We thank the reviewer for their comments. As the reviewer pointed out, if we obtain the overestimated lattice parameter of the Rh model structure in DFT calculations, the predicted chemical interaction properties of the adsorbate CO₂ molecule on the surface may have been misinterpreted. In previous studies suggested by the reviewer, Stroppa and Kresse mentioned that “*no semi-local functional is capable of describing all aspects properly, and including non-local exchange also only improves some but worsens other properties*” (Stroppa and Kresse, *New. J. Phys.* 2008, **10**, 063020.). A later study by Schimka and coworkers reported that the random phase approximation (RPA) exhibits the best performance of predicting for molecular adsorption energies (Schimka et al., *Nat. Mater.* 2010, **9**, 741-744.). Moreover, Paier and coworkers also mentioned that the RPA + exact exchange scheme can improve the general feasibility of the Kohn-Sham DFT methodology (Paier et al., *New. J. Phys.* 2012,

14, 043002.). The literature above commonly indicates that an appropriate exchange-correlation treatment is crucially important, and the use of the RPA in systematic calculations may have attained better computational prediction results than conventional DFT calculations (with GGA or hybrid functionals).

We would like to note that appropriate exchange-correlation functionals for DFT calculations on chemical reactions should consider calculation accuracy as well as computing cost. Because our constructed model (a 6×6×4 supercell structure; 144 atoms) requires a relatively large load of repeating calculations to predict the binding configurations of adsorbate CO₂ and dissociated CO molecules. We used GGA-level functionals rather than the higher level of exchange-correlation treatment for calculation efficiency. In addition, among the several conventional semi-local GGA-level exchange-correlation functionals, the RPBE suggested by Hammer, Hansen, and Nørskov can predict proper calculation results of the binding energy of the CO molecule on late-transition metals (Hammer et al., *Phys. Rev. B* 1999, **59**, 7413-7421.). In the present study, we picked out the RPBE functional in DFT calculations to understand the adsorbate CO₂ dissociation mechanism over model Rh(111) surface, and to reproduce the morphology of multiple CO bindings on the Rh surface as same as experimentally taken images by using NAP-STM technique.

Considering the points mentioned above, we further calculated the lattice parameter of Rh model structure by employing each different exchange-correlation functional, such as RPBE (3.85 Å), PW91 (3.85 Å), PBE (3.83 Å), or HSE06 (3.79 Å). It means that three conventional GGA-level functionals overestimate the lattice parameter of Rh as much as +0.03 Å (PBE) or +0.05 Å (RPBE and PW91) compared with an experimentally determined value of 3.80 Å (C. Kittel, *Introduction to Solid States Physics 8th Ed.*, Wiley & Sons, **2005**.). We would like to note that the calculated lattice parameter of Rh using RPBE functional is also overestimated in our present work, but this value has a good agreement, with experimentally confirmed results within an error of 1.3%. As discussed above, the RPBE-adopted calculations could suggest reliable binding energy values of CO molecules on late-transition metal structures in comparison with other conventional GGA-level exchange-correlation functionals (Hammer et al., *Phys. Rev. B* 1999, **59**, 7413-7421.), so we adopted the combination of RPBE functional and the D3 vdW-correction method suggested by Klimeš and Michaelides (Klimeš et al., *J. Chem. Phys.* **137**, 120901.).

Even though the higher-level exchange-correlation treatments such as the random phase approximation (RPA) (Schimka et al., *Nat. Mater.* 2010, **9**, 741-744.) or the RPA+exact exchange scheme (Paier et al., *New. J. Phys.* 2012, **14**, 043002.) may provide the more accurate surface properties of Rh model structure, our compromise predictions are still valid to explain the experimental results in direct observations of the early step of CO₂ dissociation process over Rh(111) surface. We accordingly added the following sentence in the method section of the revised manuscript.

– Page 18: Methods – DFT Calculations.

“The calculated lattice parameter of Rh was 3.85 Å (refers to Supplementary Table 4 for more discussions).”

– Supplementary Table 4.

	Experiment	HSE06	PBE	PW91	RPBE (This work)
Lattice parameter (Å)	3.80	3.79	3.83	3.85	3.85

“Lattice parameters of Rh calculated by various exchange-correlation functionals. The RPBE exchange-correlation functional is known to be overestimating the lattice parameter of late-transition metals. The calculated lattice parameter of Rh (3.85 Å) was overestimated by 0.05 Å from the experimental value. The higher-level exchange-correlation treatment methods such as the random phase approximation (RPA)¹ or the RPA+exact exchange scheme² have provided the more accurate surface properties. On the other hand, among the conventional GGA-level exchange-correlation functionals, the use of RPBE gives relatively reasonable prediction results of the binding energy of CO on late-transition metals³. Because our main purpose was understanding the dissociation mechanism of CO₂ on Rh(111) and reproducing NAP-STM images of multiple CO binding configuration, we adopted the RPBE functional and combined the D3 vdW-correction method suggested by Klimeš and Michaelides⁴ as a compromise between the calculation accuracy and the computational cost.”

– References (Supplementary Information)

- 1 Schimka, L. et al. Accurate surface and adsorption energies from many-body perturbation theory. *Nat. Mater.* **9**, 741-744 (2010).
- 2 Paier, J. et al. Assessment of correlation energies based on the random-phase approximation. *New J. Phys.* **14**, 043002 (2012).
- 3 Hammer, B., Hansen, L. B. & Nørskov, J. K. Improved adsorption energetics within density-functional theory using revised Perdew-Burke-Ernzerhof functionals. *Phys. Rev. B* **59**, 7413-7421 (1999).
- 4 Klimeš, J. & Michaelides, A. Perspective: Advances and challenges in treating van der Waals dispersion forces in density functional theory. *J. Chem. Phys.* **137**, 120901 (2012).

Reviewer comment #1-2: *With respect to CO₂ dissociation, I consider the work by Abbott and Harrison as important and related to the present work. The authors should cite J. Phys. Chem. C 2007, 111, 13137-13148 and put results into context of the previous work by Abbott and Harrison. Although, the RPBE functional was found by other workers to perform rather ok for activation barriers it still features an underestimation of barrier heights of several kcal/mol (depending on the species "transferred in the reaction") typical of semilocal (GGA-type) functionals (see e.g. Truhlar and coworkers; J. Chem. Phys. 132, 164117 (2010)).*

Response #1-2: We understand the reviewer's concern about the underestimation issue of the calculated barrier height for CO₂ dissociation over the Rh(111) surface. We have comparatively studied the literature which reported the activation energy barrier of CO₂ dissociation on Rh(111) (including the suggested previous work by Abbott and Harrison (Abbott et al., *J. Phys. Chem. C* 2007, **111**, 13137-13148.) and cited several relevant previous reports in the revised manuscript. Interestingly, we found that the activation energy barriers reported by Goodman et al. (experimental, 0.74 eV) (Goodman et al., *Surf. Sci.* 1984, **140**, L239-L243.) and Abbott et al. (theoretical, 0.79 eV) (Abbott et al., *J. Phys. Chem. C* 2007, **111**, 13137-13148.) are greater than our results (0.58 eV). Other theoretical values reported by Liu et al. (0.50 eV) (Liu et al., *J. Phys. Chem. C* 2018, **122**, 8306-8314.) and Ko et al. (0.56 eV) (Ko et al., *J. Phys. Chem. C* 2016, **120**, 3438-3447.), which were calculated with the PBE functional and the narrower supercells, are similar with our results.

These results suggest that the DFT-calculated activation energy barrier values of CO₂ dissociation on Rh(111) with the PBE or RPBE functionals are rather underestimated from the experimental value. We understand that much careful consideration is also required to evaluate the exact kinetics of CO₂ adsorption and dissociation on the Rh(111) surface. In this regard, the more advanced exchange-correlation treatment, for example, the random phase approximation (RPA) might have provided an exquisite prediction result of the CO₂ dissociation. However, considering that the supportive role of DFT calculations in this work is enough to provide a fundamental insight into the experimental findings on the CO₂ dissociation over the Rh(111) model surface. Such an underestimated activation energy barrier does not significantly disrupt overall interpretations of the CO₂ dissociation process by itself. We added more discussions to the part of results and discussion in the revised manuscript as following sentences.

– Page 15: Electronic charge analysis and proposed mechanism.

“We note that the DFT-estimated ΔE_{TS} values of CO₂ dissociation on Rh(111) with conventional generalized gradient approximation level exchange-correlation functionals are usually underestimated from the experimental value (Supplementary Table 3)⁵⁹⁻⁶². However, such underestimation does not significantly interfere with theoretical analysis of the CO₂ dissociation mechanism.”

– Supplementary Table 3.

	This work (theory)	Abbott et al. (theory)	Liu et al. (theory)	Ko et al. (theory)	Goodman et al. (experiment)
Condition	RPBE+D3	PC-MURT	PBE	PBE+D2	H ₂ /CO ₂ mixture $p(\text{CO}_2) = 1 \text{ Torr}$
E_{act} (eV)	0.58	0.79	0.50	0.56	0.74

“Previously reported activation energy barriers (ΔE_a) of the CO₂ dissociation over Rh surfaces by theoretical calculation and experiment.”

– References

- 59 Ko, J., Kim, B.-K. & Han, J. W. Density Functional Theory Study for Catalytic Activation and Dissociation of CO₂ on Bimetallic Alloy Surfaces. *J. Phys. Chem. C* **120**, 3438-3447 (2016).
- 60 Goodman, D. W., Peebles, D. E. & White, J. M. CO₂ dissociation on rhodium: Measurement of the specific rates on Rh(111). *Surf. Sci.* **140**, L239-L243 (1984).
- 61 Abbott, H. L. & Harrison, I. Activated Dissociation of CO₂ on Rh(111) and CO Oxidation Dynamics. *J. Phys. Chem. C* **111**, 13137-13148 (2007).
- 62 Liu, X., Sun, L. & Deng, W.-Q. Theoretical Investigation of CO₂ Adsorption and Dissociation on Low Index Surfaces of Transition Metals. *J. Phys. Chem. C* **122**, 8306-8314 (2018).

Reviewer comment #1-3: *The computed reaction pathway of course shows a structure corresponding to the transition state. It is required to do a normal mode analysis in order to detect the “reaction coordinate” featuring only one imaginary frequency. The authors are advised to report this mode and frequency.*

Response #1-3: We thank the reviewer for their constructive suggestion for improving our DFT calculation results. We performed a normal mode analysis and reported a single imaginary frequency of the TS. We added several sentences to the revised manuscript, as below.

– Page 19: Methods – DFT Calculations.

“The location and energy of transition state (TS) were calculated with the climbing-image nudged elastic band method^{68,69}.”

– Figure 4.

“Single imaginary frequency of 434i cm⁻¹ was calculated for the TS.”

– References

- 68 Henkelman, G., Uberuaga, B. P. & Jónsson, H. A climbing image nudged elastic band method for finding saddle points and minimum energy paths. *J. Chem. Phys.* **113**, 9901-9904 (2000).
- 69 Henkelman, G. & Jónsson, H. Improved tangent estimate in the nudged elastic band method for finding minimum energy paths and saddle points. *J. Chem. Phys.* **113**, 9978-9985 (2000).

We also provided supplementary movies for the reviewer, presenting the normal mode vibration corresponding to the imaginary frequency of the TS.

– Supplementary Movies

“NCOMMS-20-17646 Supplementary Movie R1-1.avi; The normal mode corresponding to the imaginary frequency of the TS (Side view).”

“NCOMMS-20-17646 Supplementary Movie R1-2.avi; The normal mode corresponding to the imaginary frequency of the TS (Top view).”

Reviewer comment #1-4: *The manuscript discusses of course results addressing the pressure gap of classic surface science. The calculations seem to be for a more dilute situation. The authors are advised to report the cell size or the coverage of the simulation cell for which the reaction pathway was calculated. The authors are also advised to comment on incorporating the effect of the elevated pressure in their simulations. I recommend publication once these comments have been addressed.*

Response #1-4: We agree with the reviewer’s valuable comment on the pressure gap issue in our simulation work. Indeed, the pressure gap effect is not negligible on the surface, which could be interpreted as the correlation of surface free energy and applied chemical potential at the interface. Fundamentally, the chemical potential energy is a function of the gas pressure in the closed system of various thermodynamic relations of molecular behaviors over model catalysts (e.g. a stepped {553} facet of Rh surface, and mesoscopic Rh and Pd nanoparticles), which were also investigated by Prof. Kresse and coworkers (Gustafson et al., *Phys. Rev. B* 2006, **74**, 035401.; Mittendorfer et al., *Phys. Rev. B* 2007, **76**, 233413.). We have had similar experiences figuring out the influences of the pressure gap during CO oxidation reaction over Au-CeO₂ catalysts by DFT-based investigations in recent years (Ha et al. *J. Phys. Chem. C* 2017, **121**, 26895-26902.; Ha

et al. *ACS Catal.* 2018, **8**, 11491-11501.). We have carefully considered the significance of the pressure gap effect in the present study, in which the detailed reaction steps of the CO₂ dissociation over Rh(111) surface were thoroughly investigated by the combination of advanced interface science techniques and computational predictions.

Using DFT calculations, we focused on studying CO₂ adsorption and dissociation reaction steps on the Rh(111) surface with confined molecular adsorption coverages, rather than exhaustive predictions as a function of CO₂ chemical potential, because our NAP-STM and NAP-XPS results already showed clear experimental evidence of the pressure gap effect at the CO₂-Rh(111) interface. So, we presented the calculated results of the CO₂ dissociation mechanism with the single CO₂ molecule adsorbed on a 6×6×4 Rh (111) supercell in our manuscript. Although we have not directly incorporated a computational treatment to address the pressure effect in the CO₂ adsorption and dissociation processes, our calculated single molecule CO₂ adsorption and multiple CO molecules ($\theta_{\text{CO}^*} = 0.25$ ML) binding configurations on the Rh(111) surface successfully reproduced the experimentally characterized NAP-STM images at the elevated pressures.

Interestingly, **Figure 3d** and **3e** show that the relative C2/C1 peak ratio (the degree of CO₂ dissociation) on the Rh(111) surface when exposed to 0.1 mbar CO₂ at 300 K reaches approximately 1.0 after 85 minutes, which corresponds to ~0.1 ML of dissociated CO coverage. This spectroscopic evidence suggests that the actual CO₂ coverage on the Rh(111) does not significantly exceed 0.1 ML, and chemisorbed CO₂ molecules spatially separate from each other instantaneously on the Rh(111) surface. So, our presented calculation results are valid to explain the experimentally observed CO₂ dissociation process at NAP, because the characterized CO₂ coverage is close to the ideally confined CO₂ configurations on the slab structure of Rh. However, the collision-induced behavior of CO₂ molecules at NAP should also be considered, as the effective collision and subsequent chemisorption processes may have an alteration of surface free energy at the confined surface area.

Figure R1-1. The DFT-calculated adsorption energy change between one and four CO₂ molecules configurations on a {111} facet of Rh surface. Both calculated $\Delta E_{\text{ads.}}$ values represent the average binding energy per CO₂ molecule, respectively.

To address these points, we further discuss the multiple CO₂ binding configuration on the Rh(111) surface by using DFT calculations. The predicted binding energy of single CO₂ on the Rh(111) surface (-0.39 eV) changes to -0.19 eV/CO_2 upon the adsorption of four CO₂ molecules, as illustrated in **Figure R1-1**. The proposed multiple binding configuration of CO₂ molecules has a weaker average binding energy per CO₂ molecule. It implies that a chemisorption structure with increased adsorption coverage of CO₂ molecules would exist at the higher CO₂ partial pressure environment, which provides a perspective insight of the relationship between the experimental characterization and the theoretical prediction. Considering the above discussions, we have added more descriptions of our constructed model Rh structure in the revised manuscript as below.

– Page 15: Electronic charge analysis and proposed mechanism.

“The DFT-calculated average binding energy of four CO₂ molecules adsorbed on Rh(111) was decreased to -0.19 eV/CO_2 (Supplementary Fig. 16). The multiple CO₂ binding configuration with the weaker binding energy would have a limited survival under the higher CO₂ partial pressure, providing the increased CO₂ surface coverage.”

– Supplementary Figure 16.

“The DFT-calculated adsorption energy change between one and four CO₂ molecules configurations on a {111} facet of Rh surface. Both calculated ΔE_{ads} values represent the average binding energy per CO₂ molecule, respectively.”

– **Page 18: Methods – DFT Calculations.**

“A 6×6×4 supercell was used to describe the Rh(111) surface (Supplementary Fig. 17).”

– **Page 18: Methods – DFT Calculations.**

“The mechanism of the CO₂ dissociation process over a Rh surface was studied using a model structure of single CO₂ molecule/Rh(111). Appropriate multiple molecular binding configurations were also estimated to reproduce the NAP-STM images.”

– **Supplementary Figure 17.**

“A constructed 6×6×4 supercell of the slab model of Rh in DFT calculations. A perspective projection view of the Rh model structure (**left**) and its top view image along to a direction of [111] (**right**) show the three-dimensional configurations of the Rh slab model consisting of 144 atoms.”

Response to the Reviewer #2:

General comment: *In the manuscript titled “How Rh surface breaks CO₂ molecules under ambient pressure”, the authors studied the activation of CO₂ on Rh(111) with XPS and STM under near-ambient pressure(NAP) and theoretical calculation. They evidenced the dissociation of CO₂ to form CO and O species on the Rh surface at room temperature. The observation of CO₂ dissociation with NAP techniques was studied and reported in the past several years (PNAS, 2017, 114, 266706–6711; J. Am. Chem. Soc. 2016, 138, 13246–13252; J. Am. Chem. Soc. 2016, 138, 4146–4154). For example, Eren et al detected dissociated species with NAP-XPS (J. Am. Chem. Soc. 2016, 138, 8207–8211) and Kim et al proved CO₂ dissociation using infrared reflection absorption spectroscopy (ACS Catal. 2016, 6, 1037–1044). This work does not provide much new insights on CO₂ activation although the authors obtained more information with theoretical calculation. I do not recommend its publication in Nature Communications. Some specific comments are listed in the following:*

Response: We thank the reviewer for their expert comments on our study of CO₂/Rh(111) under NAP conditions. In our manuscript, we provided atomic-scale topographic images at CO₂/Rh(111) interface and careful interpretations of adsorbate chemical binding energy analysis probed by synchrotron-based X-ray spectroscopy to explain a feasible CO₂ dissociation process over Rh catalyst at NAP. The dissociative adsorption of CO₂ molecules is a well-known early step of industrial catalytic energy conversion reactions, which affects the selectivity of the CO₂ reduction reaction (CO₂RR) during the clean gas or liquid fuel conversion process in chemical reactors. However, the atomistic activation process of CO₂ molecule at NAP was not fully understood, because only characterization under the operating condition could take an essential feature of transient geometry of CO₂ molecules (also known as *bent*-structured CO₂). Even though surface science techniques enable investigation of the unique behavior at the molecular level, most traditional analyzer designs commonly have a critical drawback for the surface analysis in reactive environments, due to the electron mean-free path issue and thermal drift problem (Ogletree et al., *Rev. Sci. Instrum.* 2002, **73**, 3872.; Tao et al., *Rev. Sci. Instrum.* 2013, **84**, 034101.; Nguyen and Tao, *Rev. Sci. Instrum.* 2016, **87**, 064101.). Recent studies with advanced NAP-design instrumental setups have shown more reliable experimental observations to help understand of CO₂RR at the interface. As

mentioned by the reviewer, the representative papers suggested clear experimental evidence of the CO₂ dissociation process at NAP. However, we would like to clarify that our research has several essential points that distinguish it from the previous literature.

First, Eren et al., (*J. Am. Chem. Soc.* 2016, **138**, 8207-8211.) showed the CO₂ dissociation process over Cu surfaces using NAP-XPS and NAP-STM. We agree that Cu is an early 3d-transition metal which is also much used in chemical engineering industries, but its surface electronic structure and adsorbate binding mechanism are definitely different to Rh catalysts. Because Rh, one of the Pt-group elements, has an electronegativity of 2.28 by the Pauling scale (Pauling, *J. Am. Chem. Soc.* 1932, **54**, 3570-3582.). Basically, detailed hybridization between the energy states of the surface band structure of Rh and the molecular orbital of CO₂ is in a distinct line compared with Cu (Electronegativity: 1.90). As a result, Rh and Cu show an enormous difference of adsorption properties when they have effective collisions with CO molecules at NAP on the facet of {111} surface, as tabulated below.

Catalyst	θ_{CO} at NAP	Reference
Rh(111)	0.75 ML	1) Present work. 2) Rider et al., J. Am. Chem. Soc. 2002, 124 , 5588-5593.
Cu(111)	~0.1 ML	1) Eren et al., Science 2016, 351 , 475-478.

Table R2-1. The characterized results of adsorbate CO coverage (θ_{CO}) on the Rh and Cu surfaces at NAP (0.05–10 mbar CO).

Second, structural sensitivity should be considered a significant requirement of the internal-bond scissoring of CO₂ molecule at NAP. In fact, the authors who suggested experimental evidence of CO₂ dissociation over Cu catalysts mentioned that the plainly flat facet of (111) is relatively ineffective in comparison with stepped facets such as {100} and {775} (Eren et al., *J. Am. Chem. Soc.* 2016, **138**, 8207-8211.; Kim et al., *ACS Catal.* 2016, **6**, 1037-1044.). In addition, the authors indicated clearly in their papers that the stepped surfaces are active for the methanol synthesis mechanism of CO₂. The view of computational predictions also suggests that the Cu catalysts have various CO₂ conversion mechanisms toward C₁ or C₂ products between the {111} and {100} facets (Peterson et al., *Energy Environ. Sci.* 2010, **3**, 1311-1315.; Garza et al., *ACS Catal.* 2018, **8**, 1490-1499.). The above reports not only provide valuable discussions of Cu catalysts, but also indicate the complexity of the selectivity, depending on the structural sensitivity of the catalyst surface. In contrast, our present work of the CO₂/Rh(111)

interface investigation shows that CO₂ molecules could be dissociated on the flat Rh(111) surface, eventually, the dissociated CO molecules from the CO₂ forms a periodic (2 × 2)-CO structure at that moment.

Third, another NAP-XPS study on the CO₂RR over polycrystal Cu catalyst (Favaro et al., *PNAS*, 2017, **114**, 6706-6711.) also extensively discussed various dissociation scenarios of CO₂ molecules. The elegant DFT calculations contributed by Prof. Goddard (Caltech) and coworkers strongly claimed that the subsurface oxygen plays a critical role as CO₂RR mechanism. On the contrary, we did not refer to the oxygen-related CO₂ conversion process in our study, so those studies provide a different insight for the CO₂RR. The other papers referred by the reviewer, which introduced the CO₂ recycling on the Ni(111) and Ni(110) (Roiaz et al., *J. Am. Chem. Soc.* 2016, **138**, 4146-4154.; Heine et al., *J. Am. Chem. Soc.* 2016, **138**, 13246-13252.), discussed important points for methanation process, but the authors mainly emphasize the formation of carbonate at NAP as an intermediate.

The three above points all suggest the keen interest and insight to understand the details of CO₂RR at the molecular level, and we have no doubts about the authors' impressive efforts on experimental investigations using NAP-XPS. However, those are in slightly different categories of surface interactions as compared with our present work. Although we have commonly used the same NAP-XPS technique, the characterized features/behaviors, facet structures, and catalyst elements are physically classified as different matters. Besides, in the referred literature by the reviewer's comment, we could not find any atom-resolved STM images at NAP together.

The novel aspect of our study is to reveal the CO₂ dissociation process on the Rh(111) surface by atomic-scale observation images for the first time. We believe our experimental data and theoretical prediction shed a light on CO₂ chemistry insights suitable for the broad interest of the readership of *Nature Communications*.

Reviewer comment #2-1: *The above mentioned previous studies represent the recent progress in CO₂ activation study and are closely related to this work. The authors may need to cite them in the manuscript.*

Response #2-1: We agree with the reviewer's suggestion related to recently published CO₂ activation studies. As we mentioned above, the authors of that literature have impressive achievements in the research community of surface science and

heterogenous catalysis. We are sure their efforts will be increasingly introduced to diverse catalysis communities, because the studies deal with the importance of detailed CO₂ activation mechanisms in more realistic reaction environments than a frozen UHV condition. The suggested papers were cited, and several sentences were added in the revised manuscript to introduce the relevant studies on the CO₂ dissociation at NAP, as below.

– Page 3: Introduction

“Recently published literature supports the strong evidence of CO₂ activation beyond the pressure gap, for instance, the intramolecular bond-breakage phenomenon of CO₂ molecules was reported on the Cu^{22,23} and Ni^{24,25} catalysts at the elevated pressures.”

– References

- 20 Favaro, M. et al. Subsurface oxide plays a critical role in CO₂ activation by Cu(111) surfaces to form chemisorbed CO₂, the first step in reduction of CO₂. *Proc. Natl. Acad. Sci. U. S. A.* **114**, 6706-6711 (2017).
- 22 Eren, B., Weatherup, R. S., Liakakos, N., Somorjai, G. A. & Salmeron, M. Dissociative Carbon Dioxide Adsorption and Morphological Changes on Cu(100) and Cu(111) at Ambient Pressures. *J. Am. Chem. Soc.* **138**, 8207-8211 (2016).
- 23 Kim, Y., Trung, T. S. B., Yang, S., Kim, S. & Lee, H. Mechanism of the Surface Hydrogen Induced Conversion of CO₂ to Methanol at Cu(111) Step Sites. *ACS Catal.* **6**, 1037-1044 (2016).
- 24 Roiaz, M. et al. Reverse Water–Gas Shift or Sabatier Methanation on Ni(110)? Stable Surface Species at Near-Ambient Pressure. *J. Am. Chem. Soc.* **138**, 4146-4154 (2016).
- 25 Heine, C., Lechner, B. A. J., Bluhm, H. & Salmeron, M. Recycling of CO₂: Probing the Chemical State of the Ni(111) Surface during the Methanation Reaction with Ambient-Pressure X-Ray Photoelectron Spectroscopy. *J. Am. Chem. Soc.* **138**, 13246-13252 (2016).

Reviewer comment #2-2: *In the abstract, the authors claimed that “Here, we report direct observations of spontaneous CO₂ dissociation over the model rhodium (Rh)*

catalyst in realistic environments.” Normally, “realistic environments” represents the reaction conditions employed in industry, for example 50-100 bar for CO₂ hydrogenation. There still exists a pressure gap between NAP and the realistic reaction condition. Therefore, it is inappropriate to use the phrase of “realistic environments” here.

Response #2-2: As pointed out by the reviewer, strictly, we did not perform experiments under harsh chemical reaction environments like an industrial chemical reactor, e.g. 50–100 bar for CO₂ hydrogenation. Our NAP-STM and NAP-XPS results reveal uncommon features during CO₂ dissociation on the Rh(111) surface compared with the previous work carried out in UHV, but that does not account for the entire physicochemical phenomenon at high pressure. Numerically, there is a surface energy difference of 0.54 eV at 300 K between 1×10^{-10} (i.e. UHV) and 0.1 mbar according to fundamental thermodynamic relations ($\Delta\mu = RT\ln P$; μ : chemical potential, R : gas constant, T : temperature, and P : pressure). The computed number is an immense value in the context of molecular interactions on the catalyst surface, probably enough to influence a certain catalytic reaction step of CO₂RR. However, the pressure gap also exists between 0.001 and 100 bar, which has a surface energy difference of 0.3 eV at least. Regarding these points, the reviewer’s criticism is reasonable, and we have changed the relevant incorrect phrase in the part of the **Abstract** on the revised manuscript, as below:

– Page 1: Abstract

“Here, we report direct observations of spontaneous CO₂ dissociation over the model rhodium (Rh) catalyst at 0.1 mbar CO₂.”

Reviewer comment #2-3: *In page 3, the authors concluded that “Here, we report direct observations of chemisorption and dissociation processes at CO₂-Rh(111) interface.” However, the authors did not provide any experimental evidences for the intermediates during the dissociation processes. The observed CO and O species can only prove the occurrence of CO₂ dissociation. In this regard, the authors did not directly observe the dissociation processes.*

Response #2-3: We appreciate the reviewer’s constructive criticism. Detailed topographic matters of direct observation evidence under 0.1 mbar CO₂ environment were not enough to explain a well-characterized process of CO₂ dissociation over

Rh(111) surface, even though we clearly proved the existence of dissociated CO and O species in **Figure 3a**. We agree that some of the wording explaining our experimental results was inaccurate and have changed the sentence pointed out by the reviewer, as below.

– **Page 3: Introduction**

“Here, we report direct observations of CO₂ molecules’ dissociative adsorption structures at CO₂-Rh(111) interface,”

Furthermore, we added a supplementary figure and description to address the direct observation evidence by the CO₂ dissociation process over the Rh(111) surface.

Figure R2-1. A real-space observation image of adsorbed CO₂ molecules’ atomistic interactions over Rh(111) surface under 0.1 mbar CO₂ environment.

Figure R2-1 shows the characterized features of chemisorbed CO₂(ads.), dissociated *t*-CO, and intermediate CO₂(ads.) together on the same topography at 0.1 mbar CO₂. The detailed CO₂ dissociation occurs in the following sequence: **i**) effective molecular collisions of the CO₂ with Rh atoms of {111} facet, **ii**) weakly-bound physisorbed CO₂(ads.) and chemisorbed CO₂(ads.) formations, **iii**) the *bent*-structured CO₂ molecules transition as an intermediate, and **iv**) dissociated CO and O formations on the surface. This picture of the molecular bonding breakage process is also in the line of provided DFT calculations in our manuscript. As shown in **Figure 3a**, the fully dissociated CO and O species could be clearly resolved on the NAP-STM image because their surface energies after the CO₂ dissociation spontaneously occupy a

relatively low-lying state, which naturally leads to the stabilized density of states on the surface. So, we could obtain the atom-resolved Rh substrate NAP-STM image as shown in **Figure 3a**. In contrast, the complex interactions between chemisorbed $\text{CO}_2(\text{ads.})$ and its transient intermediates have indeterminate ensemble structures as revealed in **Figure R2-1**.

Therefore, the additionally presented **Figure R2-1** exhibits entire forms of CO_2 -related species at the same area. We believe that the insufficient real-time observation evidence for CO_2 dissociation the reviewer pointed out can be addressed with **Supplementary Figure 12**. To our best knowledge, these type of NAP-STM images, including adsorbed *bent*- CO_2 structure and its intermediate formation under CO_2 environment, are provided in our manuscript for the first time; this would provide crucial insight for the understanding of dynamic molecular interactions between CO_2 and Rh catalyst in real space.

– Page 12: Direct observations of CO_2 dissociation on the Rh surface.

“We note that these clearly distinguished CO^* and O^* species on Figure 3a and 3b happen to be observed after the CO_2 dissociation process on the Rh(111) surface. Once the catalytic reaction initiates, the $\text{CO}_2(\text{ads.})$ molecules have complex interactions with surface Rh atoms, represented by randomly tangled structures by chemisorbed, dissociated, and intermediate forms of $\text{CO}_2(\text{ads.})$ at the same time (Supplementary Fig. 12).”

– Supplementary Figure 12.

“The complex tangled structures consisting of chemisorbed $\text{CO}_2(\text{ads.})$, dissociated *t*-CO, and intermediate $\text{CO}_2(\text{ads.})$ during the catalytic CO_2 dissociation on the Rh(111) surface. Each species indicated by the arrow with a corresponding color of green, red, or blue shows the randomly formed atomic structures, because the catalytic reaction process could make the complex interactions between adjacent adsorbate CO_2 molecules and surface Rh atoms in transition state.”

Reviewer comment #2-4: *In the NAP-STM results, the authors attributed the bright blobs in Fig. 1e to the physisorbed linear CO_2 molecules. In fact, under the condition of 0.1 mbar CO_2 , the bright blobs can be possibly caused by other impurities. The*

corresponding NAP-XPS spectra should be provided to confirm the NAP-STM assignment.

Response #2-4: The reviewer's concern is a well-known issue in the STM community, which is why many physicists have tried to observe metal or metal oxide surfaces in a measurement environment under extreme constraints, like frozen UHV conditions. There may be unmeasurable moderate amounts of the impurity within the large order of Avogadro's number gas molecules at NAP, which may disrupt the quality of image taken by wave function interferences of density of states during the flow of tunneling electrons. So, surface characterization data using STM technique should be carefully analyzed, because the ambiguous impurity can often easily lead to wrong interpretations.

In principle, the intensity of recorded tunneling current in the feedback loop of the STM system is proportional to an exponential function of z-axis separation between tip and sample in a given simplified equation from the Wentzel–Kramers–Brillouin (WKB) approximation (Jeong Y. Park, “Scanning Tunneling Microscopy”, *Characterization of Materials*, **2012**, John Wiley & Sons, Inc.). This long-standing problem could be simply resolved, if we could record tunneling current signals with impurity-free junctions on the sample. But nobody is free from such contamination, as opposed to the “*clean*” situation when investigating surfaces using modern surface science techniques. Analysis should be adapted to disentangle complexities of the surface structure. This impurity control subject has been extensively discussed from the 1980s by the experts in STM communities. It is well-known that uncertain impurity states manifest as various morphology shapes with a different contrast on the taken image, although their physicochemical identity could be classified, depending on the measured height of the protrusion and structure (H.-J. Guntherodt and R. Wiesendanger, *Scanning Tunneling Microscopy I*, **1994**, Springer-Verlag).

Following this principle, we can isolate the impurity or contaminant component on the observed three-dimensional morphologic information. The characterized appearances of “the bright blobs” are consistent with a previously published STM study of CO₂(ads.)/Ru(0001) by Prof. Salmeron group (LBNL) (Feng et al., *J. Phys. Chem. Lett.* **2015**, **6**, 1780-1784.). In addition, the progress in development of STM analysis at NAP has helped deal with the contamination/impurity issue with unceasing trial and error from the 1990s in surface science communities (Jensen et al., *J. Vac. Sci. Technol. B*, **1999**, **17**, 1080-1083.; Laegsgaard et al., *Rev. Sci. Instrum*, **2001**, **72**, 3537-3542.; Rößler et al.,

Rev. Sci. Instrum. 2005, **76**, 023705.; Tao et al., *Rev. Sci. Instrum.* 2008, **79**, 084101.; Tao et al., *Rev. Sci. Instrum.* 2013, **84**, 034101.). In this respect, we also published several works done at NAP using the most advanced version of inchworm scanner implanted Aarhus STM NAP apparatus (SPECS GmbH, Germany) in past years as part of the extended discussion about the impurity issue at NAP (Kim et al., *J. Am. Chem. Soc.* 2016, **138**, 4, 1110-1113.; Kim et al., *Sci. Adv.* 2018, **4**, eaa3151.; Kim et al., *ACS Appl. Energy Mater.* 2019, **2**, 8580-8586.). So, we believe that the NAP-STM images in the present study are correctly interpreted, regarding fundamental principles and experiences in the research field of surface science. To address the reviewer's apprehension of physisorption CO₂ evidence, we prepared another compilation of NAP-STM images, which was obtained in order of time sequence, as below.

Overlapped mapping image (Red + Green)

Figure R2-2. Atomic-scale observations of physisorption and chemisorption CO₂ molecules on the Rh(111) surface at 0.1 mbar CO₂. The representative NAP-STM image enclosed by a rectangle of red or green color (acquisition time interval; $\Delta t = 2.8$ sec) demonstrates different features of mobile and immobile bright blobs simultaneously in the overlapped mapping image, which indicates characterized distinctive features of CO₂(ads.) from the impurity at the same local area.

Figure R2-2 visualizes a different characteristic between physisorption and chemisorption CO₂ that the assigned mobile and immobile bright blobs are corresponding to physisorbed and chemisorbed CO₂(ads.) on the overlapped mapping image of NAP-STM observations. We consecutively recorded the images with a time interval of 2.8 sec at that moment, the characterized CO₂(ads.) physical parameters of lateral size (5.4 ± 0.4 Å) and height (0.3 ± 0.1 Å) are similar to previous literature (Feng et al., *J. Phys. Chem. Lett.* 2015, **6**, 1780-1784.). This real-space measurement information is absolutely far from the things appointed as an impurity (lateral size: 12 Å, height 1.0 Å) on the same place. Moreover, we can see the specified immobile bright blobs keep an ellipsoidal shape, even with a time gap in scale of seconds, which have a physical meaning of bond strength difference between chemisorption and physisorption CO₂ on the Rh(111) surface. Regarding the above discussions, we added several sentences to address the reviewer's concerns in the revised manuscript. In addition, the displayed **Figure R2-2** and its description have also been added to the revised electronic supplementary material.

– Page 6: Molecular adsorption structure observation of CO and CO₂.

“The trend of momentary diffusion of the physisorbed CO₂ molecules could be distinguished in consecutively recorded NAP-STM images with a time interval of 2.8 sec (Supplementary Fig. 3), and the characterized “immobile” chemisorption CO₂(ads.) molecules keep their ellipsoidal shape with a lateral size of 5.4 ± 0.4 Å and height of 0.3 ± 0.1 Å in real-space measurements. Moreover, the CO₂(ads.) species that appear are absolutely distinct from the impurity (lateral size: 12 Å; height 1.0 Å) on the observed local area, which could be easily isolated of the visible difference along the z-axis in three-dimensional space according to a simplified Wentzel–Kramers–Brillouin (WKB) approximation³⁵.”

– Supplementary Figure 3.

“Atomic-scale observations of physisorption and chemisorption CO₂ molecules on the Rh(111) surface at 0.1 mbar CO₂. The representative NAP-STM image enclosed by a rectangle of red or green color (acquisition time interval; $\Delta t = 2.8$ sec) demonstrates that different features of mobile and immobile bright blobs simultaneously in the overlapped mapping image, which proves characterized distinctive features of CO₂(ads.) from the impurity at the same local area.”

As pointed out by the reviewer, in principle, if the observed physisorption feature on the NAP-STM image is not an artifact, its spectroscopic analysis result should be matched each other. For instance, we can search for evidence of weakly-bound CO₂ (CO₂^{δ-}/CO₃^{δ-}) probed by NAP-XPS in literature, as below.

Model Catalyst	Peak Position (CO ₂ ^{δ-} /CO ₃ ^{δ-})	Reference
Cu(111)	288.4 eV	Baran et al., J. Am. Chem. Soc. 2016, 138 , 8207.
Ni(111)	288.5 eV	Yuan et al., ACS Catal. 2016, 6 , 4330.
Poly Cu	288.2 eV	Favaro et al., PNAS , 2017, 114 , 6706.
Cu(111)	288.4 eV	Ren et al., Chem. Eur. J. 2018, 24 , 16097.
TiO ₂ (110)	289.7 eV	Hamlyn et al., PCCP , 2018, 20 , 13122.
Ag/TiO ₂	291.9 eV	Collado et al., Nat. Commun. 2018, 9 , 4986.

Table R2-2. The previously reported peak position of weakly-bound CO₂ species (CO₂^{δ-}/CO₃^{δ-}) on various model catalyst surfaces in C 1s core-level spectra by NAP-XPS measurements.

On surveyed references, the mentioned *l*-CO₂(ads.) on our manuscript would be diversely assigned as CO₂^{δ-}, CO₃^{δ-}, physisorbed CO₂, or molecularly adsorbed CO₂ with varied binding energy between 288.2–291.9 eV in C 1s core-level spectra, depending on the analyzed substrate in NAP-XPS measurements. Even though our searched information cannot cover all circumstances of the weakly-bound property of CO₂ molecules on the surface at NAP, a difference of 3.7 eV between lowest and highest values is a large number to absolutely define a specific character. The surface interaction between adsorbates and exposed facets is affected by the electronic structure of the substrate, so the adsorption property of molecular behavior could have

been influenced with the complex hybridizations between substrate metal atoms *d*-band and molecules *s*-/*p*-orbital structures on the surface.

In our study, we found two signature peaks between 285–287 eV except a gas-phase peak in C 1s core-level spectra under gaseous CO₂ environment, but some notable peak by naked eyes not appeared simultaneously, as shown in **Figure 2b**. Instead, we have presented a supplementary figure, as below, which is more sensitive to the adsorbate species than the NAP-XP spectrum in **Figure 2b** at a different distance of sample-to-aperture for the NAP-XPS analysis.

Figure R2-3. Acquired C 1s NAP-XP spectrum at 0.1 mbar CO₂ after gas exposure of 8 min at a different distance of sample-to-aperture.

Unlike **Figure 2b**, we could additionally confirm a small peak at a binding energy of 289.4 eV by an adjustment of sample-to-aperture distance (Amann et al., *Rev. Sci. Instrum.* 2019, **90**, 103102.). It does not mean the change of backfilled pressure, nor the spectral artifact. Because the photo-emitted electrons from the sample are scattered by gas molecules between the sample and cone of the hemispherical analyzer, which can generate a different gas phase to sample surface peak ratio in NAP-XPS measurements. It originated from the effective gas collision distribution near a hole of the aperture nozzle, *i.e.* the specific yield of photoelectrons in the gas environment depends on the geometrical distance between the nozzle and the sample surface. Regarding the principle, obtained features in **Figure R2-3** implicate the more adsorbate-sensitive analysis result than that of **Figure 2b**. Thereby, the characterized weakly-bound

CO₂(ads.) species at 289.4 eV in C 1s core-level spectrum would be described as the physisorption *I*-CO₂ molecules in real-space observations. Considering the above discussions, we have polished the paragraph on C 1s core-level spectra analysis under a 0.1 mbar CO₂ environment in the revised manuscript. In addition, we have added the spectroscopic evidence of weakly-bound CO₂(ads.), which is consistent with our NAP-STM image, as below.

– Page 9: X-ray spectroscopy analysis of chemical binding energy.

“Interestingly, two peaks of the assigned CO₂(ads.) and CO(dis.) were identified at 285.5 and 286.1 eV. They are corresponding to chemisorbed CO₂ and dissociated CO adsorbates in C 1s core-level spectrum, which have the same deconvolution profiles as at full width at half-maximum (FWHM) and peak center positions at the binding energy for the analysis results of CO/Rh(111).”

“However, the effective collisions of CO₂ free molecules on the Rh(111) surface would form weakly bound CO₂(ads.) species^{20,22}, as observed in Figure 1e, which differs remarkably from the measured NAP-XPS results in the CO(g) environment. This unique property is characterized at the early stage of gaseous CO₂ exposure, the assigned peak of CO₂^{δ-} at 289.4 eV in C 1s core-level spectrum (Supplementary Fig. 8) suggests a clear evidence of molecular CO₂ interactions with Rh atoms at NAP.”

– Supplementary Figure 8.

“Adsorbate-sensitive C 1s NAP-XP spectrum under 0.1 mbar CO₂(g) environment at a different distance of sample-to-aperture. This spectroscopic result was measured after the gas exposure of 8 min at 300 K. The assigned peaks of CO₂(ads.), CO₂^{δ-}, and CO₂(g) are corresponding to chemisorption, physisorption, and scattered gas molecule species. Contrary to Figure 2b, the CO₂(ads.) species for chemisorption is also assigned at 286.1 eV instead of the CO(dis.), because the CO₂ dissociation process is not yet fully triggered at that time.”

Reviewer comment #2-5: *In the NAP-XPS results, Fig. 2a shows the pressure of 0.1 Torr whereas in the manuscript, it was described that the experiments were carried out*

at 0.1 mbar CO_2 and 0.1 mbar CO . Besides, the Rh core level spectra in CO_2 and CO environments only showed the fitted green peak. However, it can be clearly found that there exist other intensities at the left side of the green peak. The author may want to provide some discussions.

Response #2-5: We appreciate the reviewer's careful reading of our manuscript. We have corrected the typo of "0.1 Torr" as "0.1 mbar" in **Figure 2a** on the revised manuscript. We agree with the reviewer's indication of our interpretation on the fitted green peak that each NAP-XP spectrum has a slightly broader peak-width rather than the measured "clean" spectrum in UHV. We provide additional analysis results as shown in **Figure R2-4** to address the influence of adsorbates for Rh 3d core-level spectra in gas environments.

Figure R2-4. Collected Rh 3d core-level spectra in 0.1 mbar CO or CO_2 environments. Each evolved Rh peak is compared with the "Clean" Rh surface spectrum.

In **Figure R2-4**, we cannot find a surface state of Rh after gas exposures at elevated pressures for which each separated figure exhibits the peak-width broadening phenomenon for the Rh bulk peak in common. Also, we can see comparative differences assigned as "Adsorbates" on overlapping comparison results of spectra in 0.1 mbar CO and CO_2 condition. It is well-known that the Rh 3d core-level peaks are affected by local adsorbate configurations on the surface. So, the measured surface core-level would be shifted depending on the site occupation of adsorbed molecules, and the adsorbate uptake properties were carefully investigated on Rh surfaces in previous reports

(Andersen et al., *Phys. Rev. B* 1994, **50**, 17525.; Weststrate et al., *Surf. Sci.* 2004, **566**, 486-491.; Vesselli et al., *Phys. Rev.* 2004, **70**, 115404.; Bianchettin et al., *J. Phys. Chem. C* 2007, **111**, 4003-4013.; Vesselli et al., *J. Phys. Chem. C* 2008, **112**, 14475-14480.; Bianchettin et al., *J. Phys. Chem.* 2009, **113**, 13192-13198.; Toyoshima et al., *J. Phys. Chem. C* 2015, **119**, 3033-3039.).

As mentioned, the investigated surface Rh 3d core-level shifts of H/Rh and S/Rh in the literature are involved in molecular adsorption-induced changes of core-level electrons binding energy and electronic charge redistributions by the bonding formation of adsorbates–Rh surface. In the same manner, we indicate the CO adsorption feature with arrows in **Figure R2-4**, and these results are consistent with the characterized Rh 3d core-level spectra under CO pressure of 10^{-7} and 50 mTorr using the same NAP-XPS setup (Ueda et al., *ACS Catal.* 2018, **8**, 11663-11670.). We agree that the specific molecular adsorption sites on the Rh surface would have been assigned by careful peak deconvolution methods on acquired Rh 3d core-level spectra, but we should consider the presence of possible contributed peaks from the gas molecules scattering phenomenon in NAP condition. Actually, this feature is well-understood with respect to the inelastic electron scattering process by gas phase molecules, for which a remarkable and recently developed “high-pressure XPS” facility at DESY (Germany) has highlighted the signature as an enhanced spectral tail broadness at the elevated pressures up to few bars (Amann et al., *Rev. Sci. Instrum.* 2019, **90**, 103102.).

So, although we may attribute the influence of adsorbates on the Rh surface to an ensemble in NAP condition, we could not define site-dependent bonding properties between adsorbates and Rh(111) surface at 0.1 mbar CO and CO₂. We have discussed the above indicated points of adsorbates-induced surface core-level shift and photo-emitted electrons inelastic scattering at NAP more thoroughly in the revised manuscript, as indicated below.

– Page 8: X-ray spectroscopy analysis of chemical binding energy.

“Their delicate spectral changes, depending on the different kinds of adsorbate gas molecules (i.e. CO or CO₂) at NAP is analyzed in the comparison plot of Rh 3d core-level spectra taken in each different gaseous environment (Supplementary Fig. 7). The adsorption of gas molecules and their effective collision behaviors on the Rh(111) surface commonly make a noticeable broadening of the peak at 307.5 eV beside the characterized portion of Rh bulk

species in the Rh 3d core-level spectra. In particular, we can find a small spectral shoulder at 307.9 eV, as displayed in the overlapping comparison plot of CO(g) and CO₂(g) in Supplementary Fig. 7, which implies that the surface Rh 3d core-level shifts get involved in the adsorbate–Rh atoms bonding formation properties by reactive molecule collisions and electronic charge redistribution on the Rh(111) surface⁴⁴⁻⁴⁶.”

– References

- 44 Bianchettin, L. et al. Experimental and Theoretical Surface Core Level Shift Study of the S-Rh(100) Local Environment. *J. Phys. Chem. C* **111**, 4003-4013 (2007).
- 45 Bianchettin, L. et al. Surface Core Level Shift: High Sensitive Probe to Oxygen-Induced Reconstruction of Rh(100). *J. Phys. Chem. C* **113**, 13192-13198 (2009).
- 46 Amann, P. et al. A high-pressure x-ray photoelectron spectroscopy instrument for studies of industrially relevant catalytic reactions at pressures of several bars. *Rev. Sci. Instrum.* **90**, 103102 (2019).

Reviewer comment #2-6: *In Fig 2b and 2c, both the C1s and O1s spectra show the chemisorption of t-CO and h-CO at 0.1 mbar. In principle, the signal intensity ratio of t-CO/h-CO in C1s should be the same with that obtained from O1s with the same gas exposure. As indicated by the authors, the relative C 1s signal intensity ratio of t-CO/h-CO is about 0.5, while it is close to 1 from O1s results. The authors may provide the calculated t-CO/h-CO ratio for O1s and give the explanation?*

Response #2-6: The reviewer’s expert view about XPS analysis is correct, and the signal intensity ratio discrepancy between C 1s and O 1s core-level spectra at 0.1 mbar CO and CO₂ in the present study should be addressed. To conclude, it is hard to say that we can constantly collect similar peak intensity ratios between C 1s and O 1s due to the relationship of kinetic energy and diffraction of photoelectrons on the surface. Even though populated photoelectrons result from adsorbate site occupations on the surface, the involved molecular geometry could make a measurable photoelectron diffraction varied with a function of photon energy.

For CO/Rh(111) study, a reported representative study critically indicated (Beutler et al., *Surf. Sci.* 1997, **371**, 381-389.) the controversial spectral interpretation between C 1s and O 1s core-level spectra (Delouise et al., *Surf. Sci.* 1984, **147**, 252-262.); *i.e.* the authors emphasized the influence of photoelectron diffraction effect in the analysis of site-specific CO adsorbates structure on the Rh(111) surface. In the case of energy scanned X-ray photoelectron spectroscopy, the strength of kinetic energy dependency can make a significant variation of different peak intensity ratio. The C 1s core-level spectra of ethylidyne/Rh(111) shows the result of an almost completely suppressed picture of higher binding energy components by the diffraction effect (Wiklund et al., *Surf. Sci.* 2000, **461**, 107-117.). The trend that the site specific adsorbate coverage at on-top, bridge, and hollow have relatively different intensities of Pt 4f surface components is also found in the characterization of adsorbate overlayer structures on the Pt(111) surface. However, the intensity of peaks of each occupation have absolutely no match.

In the present study, the relative signal intensity ratio of *t*-CO/*h*-CO in C 1s core-level spectrum at 0.1 mbar CO has 0.51, but this value does not correspond with the O 1s core-level spectrum of 0.80. In the same manner, the numerical peak ratio of blue/magenta color for C 1s NAP-XPS spectrum at 0.1 mbar CO₂ is 1.21, whereas the taken O 1s core-level spectrum in same condition has a calculated value of 1.68. Our investigated NAP-XPS results only indicate an opposing tendency of signal intensity ratio between CO and CO₂ gas environments, which cannot explain the quantitative interpretation of adsorbate coverage dependency, according to the literature. Furthermore, it is significant that the molecular adsorption geometry of CO₂ has a similar appearance to CO adsorbates when the intramolecular CO bonding part of CO₂ takes into the atop site of the Rh surface while the other side of the bonding has no contact to Rh atoms. Further steps of CO₂ dissociation only occur when the *bent*-structured CO₂ adsorbate gets into transition state by the inequality of charge transfers between CO₂ molecules and the Rh surface. Then, the O 1s core-level spectrum at 0.1 mbar CO₂ shows the evolved intermediate peaks of resolved O1 and O2 in contrast with the C 1s spectra at equilibrium. We note that the bonding geometry of *bent*-CO₂ molecule on the Rh(111) surface is quite distinct from specified spectral characters at 0.1 mbar CO.

Thereby, the collected C 1s and O 1s core-level spectra at 0.1 mbar CO₂ may show slightly different bonding geometry species depending on the progress of CO₂ dissociation, whereas acquired NAP-XPS results at 0.1 mbar CO do not show the

unexpected tendency, because adsorbate CO molecules have only a direct chemical binding between the C atom of the CO molecule and the atop or hollow site of the Rh surface at a fixed geometry. To address the reviewer's point, we have revised and added several sentences to explain the details of the different peak intensity ratio issue, as below.

– Page 8: X-ray spectroscopy analysis of chemical binding energy.

“Moreover, the relative C 1s signal intensity ratio (Supplementary Table 1) of *t*-CO/*h*-CO is 0.51, and this spectroscopic evidence supports the topographic observation result of (2 × 2)-3CO structure in Fig 1d. However, the characterized ratio of *t*-CO/*h*-CO in the C 1s core-level spectrum is not matched with the calculated ratio in the O 1s core-level spectrum (0.80). Because the collected signal intensities would be influenced by the kinetic energy of irradiated photon beams during the capturing of photo-emitted electrons at the interface. It does not mean the qualitative change of characterized adsorbate species during the NAP-XPS measurements. The populated photoelectrons could be measured with different signal intensity ratios of *t*-CO/*h*-CO between C 1s and O 1s core-level spectra by the influence of adsorbate geometry and photoelectron diffraction effect as a function of photon energy, as reported previous literature^{40,48,49}.”

– References

- 40 Beutler, A. et al. On the adsorption sites for CO on the Rh(111) single crystal surface. *Surf. Sci.* **371**, 381-389 (1997).
- 48 DeLouise, L. A., White, E. J. & Winograd, N. Characterization of CO binding sites on Rh{111} and Rh{331} surfaces by XPS and LEED: Comparison to EELS results. *Surf. Sci.* **147**, 252-262 (1984).
- 49 Wiklund, M., Beutler, A., Nyholm, R. & Andersen, J. N. Vibrational analysis of the C 1s photoemission spectra from pure ethylidyne and ethylidyne coadsorbed with carbon monoxide on Rh(111). *Surf. Sci.* **461**, 107-117 (2000).

– Supplementary Table 1.

NAP-XPS Gas environment	Chemical Equilibrium	C 1s Peak Intensity ratio	O 1s Peak Intensity ratio
0.1 mbar CO	Equilibrium	0.51 (t -CO/ h -CO)	0.80 (t -CO/ h -CO)
0.1 mbar CO ₂	Initial	0.70 (C2/C1)	1.05 (O3/O2)
	Equilibrium	1.21 (C2/C1)	1.68 (O3/O2)

“Comparison of relative NAP-XPS signal intensities ratio of deconvoluted peaks obtained in 0.1 mbar CO and CO₂ conditions.”

Reviewer comment #2-7: *To elucidate the formation of O1 in Fig. 2c, the authors sequentially acquired O 1s core-level with different gas exposure (Supplementary Figure 5). They reported the peak shifts of O1 and O2 from their fitting results. However, the peak positions of O1 and O2 are very close to each other. Thus, the fitted results for peak shift could be artificial. It would be more reliable if the authors refer to the corresponding peaks ratios in C1s for O1s peaks analysis.*

Response #2-7: We understand that the reviewer has doubts about our NAP-XPS interpretations by the fitting method. We recognize that inconsistent peak deconvolution procedures may lead to misinterpretations of NAP-XPS data. To exclude this issue, we strictly kept our peak deconvolution procedure, using constant FWHM of each assigned species, and used a widely accepted function of mixed Gaussian–Lorentzian (70%:30%) as described in our manuscript and ESM. The number of characterized adsorbate peaks is just two or three for each C 1s and O 1s spectra in **Figure 2**, which would be considered a well-acceptable XPS peak fitting method in the traditional surface science community. We present an additional figure of O 1s core-level spectra analysis to address the peak deconvolution procedure indicated by the reviewer, as below.

Figure R2-5. The comparison of initial ($t_0 + 14$ min) and equilibrium ($t_0 + 132$ min) O 1s core-level spectra under 0.1 mbar CO₂ environment.

Figure R2-5 clearly shows a spectral broadening feature between green and red color spectra, which means that the O 1s core-level spectrum at equilibrium ($t_0 + 132$ min) may have another peak (an assignment of “O1” in the manuscript) compared with the initial one ($t_0 + 14$ min) in gaseous CO₂ condition. We explained the change of O 1s NAP-XPS spectra by an intermediate evolution during the CO₂ dissociation in time-lapse measurements; there is a difference of 0.5–0.6 eV for deconvolution center of peaks between “O1” and “O2”. Previously published XPS analysis papers, including the literature suggested by the reviewer, show even a narrow peak position difference of 0.1–0.2 eV with various decompositions for at least five species in NAP-XPS spectra (Roiaz et al., *J. Am. Chem. Soc.* 2016, **138**, 4146-4154.; Favaro et al., *PNAS*, 2017, **114**, 6706-6711.).

For the analysis of NAP-XPS data under a gaseous CO₂ environment, we separated two states of initial and equilibrium in order of time sequence. We could not observe remarkable spectral shape changes of C 1s and O 1s core-level spectra for an extended duration under CO environment, because the CO adsorbates occupy top and hollow sites of Rh(111) surface with a coverage of approximately 0.75 at 0.1 mbar CO. In contrast, the adsorbate interactions of CO₂ over the Rh(111) surface are accompanied

by diverse steps of adsorption and intermediate transition during CO₂ dissociation. So, we need to confirm the reliability of peak evolution in a clear-cut distinction between initial and equilibrium spectra in NAP-XPS measurements. As shown in **Supplementary Table 1**, C 1s peak intensity ratio (C2/C1) for 0.1 mbar CO₂ environment increased 172.9% between initial (0.70) and equilibrium (1.21), and its corresponding O 1s peak intensity ratio (O3/O2) also increased 160.0%. This implies that some dissociate CO contributes to the relative peak intensity ratio at equilibrium, which indicates that our core-level spectra decomposition analysis results are consistent with the atom-resolved NAP-STM images in real-space, and that CO₂ adsorbates interact with Rh surface atoms slowly via the catalytic dissociation pathway. We have added more explanation to the revised manuscript, as below.

– Page 10: X-ray spectroscopy analysis of chemical binding energy.

“We can find a clear spectral broadening feature between initial ($t_0 + 14$ min) and equilibrium ($t_0 + 132$ min) O 1s core-level spectra in time-lapse NAP-XPS measurements, and the deconvoluted O1 peak is distinct from an adjacent O2 species with binding energy differences of 0.5–0.6 eV (Supplementary Fig. 10).”

– Supplementary Figure 10.

“The comparison of initial ($t_0 + 14$ min) and equilibrium ($t_0 + 132$ min) O 1s core-level spectra under 0.1 mbar CO₂ environment. The plotted spectra show a remarkable difference of signal intensity at around 530.5 eV, because there is a significant spectral broadening feature. The measured peak-to-peak difference of binding energy between O1 and O2; i.e. $\Delta(O1,O2)$, is 0.5–0.6 eV, which provides evidence of intermediate species (O1) evolution by CO₂ dissociation process at equilibrium.”

– Page 10: X-ray spectroscopy analysis of chemical binding energy.

“As a result, the measured peak intensity ratio (C2/C1) in C 1s core-level spectra under 0.1 mbar CO₂ environment increase 172.9% between initiation and equilibrium (Supplementary Table 1), because the spectral portion of dissociated CO contributes to the change of relative signal intensity ratio. The corresponding relative peak intensity ratio of O3/O2 in O 1s core-level spectra of initial and equilibrium also increased 160.0%, indicating that the chemical

species interpretation using a widely used peak deconvolution procedure for NAP-XP spectra are thereby reliable as supporting evidence of the CO₂ dissociation process.”

Reviewer comment #2-8: *In Fig. 4d, the peak of C1 was assigned to chemisorbed CO₂, but the STM results in Fig. 4a and b do not show CO₂ chemisorption. Although the authors observed the ordered structure in Supplementary Figure 2, it could be caused by the variation of STM tip state. The authors may need to provide other evidence for chemisorbed CO₂.*

Response #2-8: We addressed the reviewer’s indicated point of evidence of chemisorbed CO₂ on NAP-STM images with **Response #2-4**. We agree with the reviewer’s comment that tip-induced artifacts of evolution might occur during the scanning of the sample surface. However, our STM operation procedures, gas impurity controls, and recorded image analysis were strictly handled, as mentioned in **Response #2-4** and the manuscript. To be sure of our arguments in this response, we present another NAP-STM image, which was taken at a different tunneling condition below.

$V_t = 0.67 \text{ V}; I_t = 0.16 \text{ nA}$

Figure R2-6. A recorded NAP-STM image in 0.1 mbar CO₂ at a different tunneling condition ($V_t = 0.67\text{V}; I_t = 0.16 \text{ nA}$).

Figure R2-6 shows the captured mobile CO₂(ads.) during NAP-STM measurements. The applied sample bias was 0.67 V, which has a different tip state for tunneling electrons flow compared with the other tunneling conditions of 0.22 V and 0.54

V. Even though we measured the adsorbate CO₂ before getting into the intermediate transition (**Figure R2-1**) at different tip states, the three represented images show a similar ellipsoidal shape of CO₂ molecule. The measured shape and morphology are supposed to change at each different tunneling condition, if the specific matter on STM images has a completely dissimilar density of states or uncertain artifacts.

However, we consistently taken similar NAP-STM images in 0.1 mbar CO₂ environments, as discussed above. Overall, our arguments are not only in agreement with the traditional consensus of surface science community but also reliable in comparison with other reported NAP studies. A detailed explanation and supplementary figure have been added to the revised manuscript. Our discussion and verified analysis results should sufficiently address the reviewer's comments.

– Page 6: Molecular adsorption structure observations of CO and CO₂.

“In addition, the characterized CO₂(ads.) molecules are consistently observed at different tunneling conditions in NAP-STM measurements, showing that the recorded tunneling images of CO₂–Rh interface analysis results are far from a tip-induced artifact in direct observations (Supplementary Fig. 4).”

– Supplementary Figure 4.

“Representative line profile analysis results of the characterized CO₂(ads.) molecule at different tunneling conditions of 0.22, 0.54, and 0.67 V at 0.1 mbar CO₂ environment, which indicates that the recorded NAP-STM images of

CO₂(ads.) molecules are irrelevant to the STM tip-induced artifact between 0.22 and 0.67 V of positively biased voltage.”

Response to the Reviewer #3:

General comment: *This paper by Kim et al. presents a novel study of the interaction of moderate pressures of CO₂ with an active Rh surface using advanced near-ambient pressure microscopy and spectroscopy. The subject matter is of keen interest both to the traditional surface science community and also to a wider audience with its clear relevance to catalysis. There are, however, a few matters that need to be addressed in my opinion before this manuscript is ready for publication, these are detailed below:*

Response: We appreciate the constructive criticism of our interface science study between CO₂ and the Rh(111) surface under near-ambient pressure conditions. As mentioned in the reviewer's comment, the subject of CO₂/Rh(111) interface has been of interest in the research areas of surface science and heterogenous catalysis from 1970s, particularly, Prof. Somorjai (UC Berkeley) and Prof. Weinberg (Caltech) debated intensely on the possibility of CO₂ dissociation over Rh catalysts at atmospheric pressure (Sexton et al., *J. Catal.* 1977, **46**, 167-189.; Weinberg, *Surf. Sci. Lett.* 1983, **128**, L224-L230.). The conflicting opinions had fairly reasonable points on their own experimental and theoretical investigations of each other. Unfortunately, traditional surface science techniques have had a critical limitation when looking into the details of gas-solid interface in ambient pressure environments, owing to the inelastic electron mean-free path issue; as a result, traditional surface characterizations of the early molecular activation steps for the CO₂ reduction reaction have left gaps in our knowledge over the past 40 years. To address this controversial story, we carefully focused on investigations of CO₂ molecule interactions over the Rh(111) model catalyst at the interface in moderate pressure conditions. The collected direct observations results with employing NAP-STM and NAP-XPS techniques decisively show evidence of catalytic CO₂ dissociation without the aid of hydrogen molecules. We believe that our works provide a clear answer to the previous question raised between two prominent groups in the heterogenous catalysis community, which may be significant to the readership in *Nature Communications*.

Reviewer comment #3-1: *The issue of photon-induced reactions/cracking is raised by the authors, but is not clearly ruled out in their data, which could have consequences for the conclusions they draw. High flux density soft X-rays, especially those generated by*

modern insertion device beamlines are known to routinely lead to cracking of background CO inside UHV chambers leading to carbon build-up on the sample under investigation, along with other deleterious changes. Such beam-induced modifications are even more apparent in ambient pressure XPS, and I think the authors need to demonstrate thoroughly through a suitable control study that such effects are not the cause of the changes they are seeing. One relatively simple study would be to repeat the high pressure CO₂ long exposure experiment in the absence of the photon beam to ensure that the same result is gained.

Response #3-1: We thank the reviewer for giving us their expert critique on our NAP-XPS results. Based on our past years of experiences with the soft X-ray beamline, we understand the reviewer's reasonable suspicion of background carbon contamination during NAP-XPS measurements. In addition, the radiation-induced carbon contamination issue has been well-known enough that some beamline scientists had already defined them as a major problem when they had X-ray photon-induced experiments with high-flux beam intensity by employing insertion devices in generating intense synchrotron radiation. For example, a representative report by soft X-ray synchrotron radiation scientists at the TEMPO beamline at the SOLEIL in France showed the quantitative changes of mass fragments corresponding to C, CO, and CO₂ with and without synchrotron radiation beam in the first mirror chamber (Chauvet et al., *J. Synchrotron Rad.* 2011, **18**, 761-764.). In the traced mass spectrometry results, we find that the high-flux photon beam makes a difference of 1–2 orders of magnitude for each carbon-related fragment, in comparison with measured data in base vacuum. As the result, the photon-induced dissociated carbon species are gradually accumulated on the optics devices inside the mirror chamber, which hampers the acquisition of reliable photon flux in transmission curve at beamline operation periods.

As pointed out by the reviewer, this problem routinely occurs at soft X-ray beamline facilities. Specifically, a large volume of gas experiments at NAP may have accelerated this problem more than UHV-based measurements over a short time. Such a photodissociation process would be closely related to part of the photoionization phenomenon in atomic–molecular physics; in particular, the cross-section of photo-ion yield acquired in the region between VUV and XUV indicates that some strong resonance vibrational peaks of fine structure are assigned to the Tanaka–Ogawa and Rydberg series with experimentally defined quantum defect values (McCulloh, *J. Chem.*

Phys. 1973, **59**, 4250-4259.). The intermittently observed resonance structures of CO_2^+ were also revealed by ZEKE photoelectron spectroscopy, using a coherent XUV laser source (Fielding et al., *Chem. Phys.* 1991, **155**, 257-265.); the authors reported the pre-dissociation feature in the region of the CO_2^+ ($^2\Pi_{g3/2}$) ionization limit. We can describe this phenomenon as a given equation of “ $\text{CO}_2 + h\nu$ (VUV) \rightarrow $\text{CO} + \text{O}$ ”, and it is widely accepted with photo-fragment evolutions that the dissociation pathway on the potential energy surface of the CO_2 makes a photo-fragment of C (^3P) from the electronically excited *bent*-geometry of the CO_2 (Lu et al., *Science*, 2014, **346**, 61-64.). Thus, we can assume that the photon-induced final product in the gaseous CO_2 environment at the beamline facility is an atomic carbon fragment.

As the matter of fact, we could not observe the atomic carbon or oxygen in the collected NAP-XPS results during repeated experiments at NAP. It is correct that the Rh(111) sample was exposed for over 1 hr under gas environments, as shown in **Figure 2a-c**, but that does not mean a long exposure of high-density photon beams together at that moment. The duration of actual photon beam exposure was within 3 min at a shot of core-level spectrum acquisition, and we had 12 total shots of beam exposure to investigate C 1s and O 1s core-level spectra sequentially. So, we shut off the beam shutter when we adjusted gap of insertion devices to get different photon energy with an optimizing mirror position so that the backfilled gas and Rh(111) sample were not continuously irradiated by the strong photon beam during NAP-XPS experiments. The plotted time-lapse measurements of C 1s core-level spectra at 0.1 mbar CO_2 in **Figure 3d** support that the peak of dissociated CO^* from CO_2 (assigned as “C2” in the plot) was not linearly increasing as a function of lapsed time. Therefore, the dissociated CO is a product via a catalytic reaction mechanism on the Rh(111) surface, and intermediate formations from $\text{CO}_2(\text{ads.})$ are involved in this complex phenomenon. We present more evidence below to support our arguments by comparing the spectral plots of before and after pump down.

Figure R3-1. C 1s core-level NAP-XPS measurements at 0.1 mbar CO and CO₂ and after pump down.

Figure R3-1 displays the remaining adsorbate evidence after long exposure of specific gas molecules in the NAP-XPS analysis chamber. At 0.1 mbar CO, we can see the *t*-CO and *h*-CO (red color) with a signal intensity ratio of 0.51 (**Supplementary Table 1** in the revised ESM); the portion of *h*-CO drastically decreased (black color) in the plotted C 1s core-level spectra after pump down. It is similar to previously reported analysis results of CO/Rh(111) at 10^{-7} mTorr (Ueda et al., *ACS Catal.* 2018, **8**, 11663-11670.), which imply that the remaining CO coverage after evacuation is about 0.25–0.33 ML. In contrast, we can see the dissociated CO and intermediate CO₂ peaks (black) after CO₂ gas evacuation, and the interesting trend of compared signal intensity ratio between black- and red-colored spectra is opposed to the measured result at 0.1 mbar CO.

Similar experimental investigation of CO₂ dissociation was approached on the Cu(997) surface at the endstation of the BL07LSU beamline at Spring-8 (Japan), the authors reported that they acquired almost the same NAP-XP spectra in a dark and beam-induced environment with long-term CO₂ gas exposure over 1 hr (Koitaya et al., *Top. Catal.* 2016, **59**, 526-531.). This indicates that the contamination or impurity issue may occur according to the fundamental principle, but their effective influence may vary, depending on the detailed optics and chamber configurations, synchrotron beam intensity, pumping speed, and irradiation time.

In our case, we had already excluded the unwanted issues, such as beam-induced photo-fragments evolutions and gas impurity diffusions, before core-level spectrum acquisitions at NAP by blank XPS measurements and monitoring fragments of m/z 12 and 28 corresponding to atomic carbon and CO with installed mass spectrometer at the first differential pumping stage. Furthermore, we intentionally avoided a potential issue with X-ray beam-induced sample damage, as shown in **Supplementary Fig. 9**. We believe our NAP-XPS signals were thoroughly collected with excluded contamination issues at NAP as much as possible. We added more discussions related to the X-ray beam-induced contamination issue in the revised manuscript, as below.

– Page 18: Method - Synchrotron-based NAP-XPS experiments.

“The high-flux photon beam was irradiated to the Rh(111) model catalyst within 3 min at a shot of the selected core-level analysis, then the beam shutter was closed immediately after the acquisition of each spectrum.”

– Supplementary Table 1.

NAP-XPS Gas environment	Chemical Equilibrium	C 1s Peak Intensity ratio	O 1s Peak Intensity ratio
0.1 mbar CO	Equilibrium	0.51 (t -CO/ h -CO)	0.80 (t -CO/ h -CO)
0.1 mbar CO ₂	Initial	0.70 (C2/C1)	1.05 (O3/O2)
	Equilibrium	1.21 (C2/C1)	1.68 (O3/O2)

“Comparison of relative NAP-XPS signal intensities ratio of deconvoluted peaks obtained in 0.1 mbar CO and CO₂ conditions.”

– Page 11: X-ray spectroscopy analysis of chemical binding energy.

“We emphasize that the represented spectroscopic evidence of dissociated CO* from CO₂(ads.) was obtained in the strictly managed X-ray photoemission experimental setup to exclude the issue of photon-induced contaminations⁵². The high-flux X-ray photon beam was not continuously irradiated to the Rh(111) single crystal proportional to the exposure time of CO₂ gas molecules in the analysis chamber. No significant evolution of carbon fragment or carbidic

species was found in the C 1s core-level spectra at NAP conditions, which is also confirmed in the plotted comparison spectra of before and after pump down (Supplementary Fig. 11)."

– Supplementary Figure 11.

"Acquired C 1s core-level NAP-XPS measurements ($h\nu = 400$ eV) at the Rh(111) interface under 0.1 mbar CO and CO₂ (red color), and after pump down (black color). The comparably plotted reactant gas-in and gas-out spectra in each different environment demonstrate the change of adsorbate coverage on adsorption sites consisting of Rh atoms. Representative photo-induced contaminants, such as atomic carbon and carbidic species, did not appear during the NAP-XPS analysis."

Reviewer comment #3-2: *The gas feed purity is also essential in ambient pressure measurements as even trace impurities can cause misleading results, please could the authors explain in detail the steps they took to ensure that the gas feed was clean.*

Response #3-2: We agree with the reviewer's concern about controversial gas impurities during surface characterizations under ambient pressure conditions (Trotochaud et al., *J. Phys. Chem. B* 2018, **122**, 1000-1008.; Eren et al., *Phys. Chem. Chem. Phys.* 2020, in press.). We are well aware that trace contaminants could distort experimental analysis, in particular, such as carbon, oxygen, metal-carbonyl [Ni(CO)_n or Fe(CO)_n], and water impurities mainly affect collected NAP-XPS spectra. To remove gas impurities, we had further purification procedures using genuine filters or cold traps in a gas manifold system. All connected gas lines to feed the precision leakage valve of the analysis chamber had a bake out procedure at 110°C with a high-speed pump out by a turbo molecular pump at least 12 hr before surface measurements in NAP conditions.

Another important point is gas delivery design of a manifold in the system that we minimized each assembly of a length within 15 inches from the specific gas cylinder as much as possible. For the gaseous CO₂ feed, we used high purity (99.995%) CO₂ filled with a gas cylinder at the endstation of the BL-13B beamline at PF-KEK that the CO₂ gas was further purified by repeated cycles of the freeze-pump-thaw method using a cold trap to remove unwanted impurities in the gas cylinder. This procedure is effective

for obtaining contaminant-free NAP-XP spectra at the beamline facility, which is also used by other NAP-XPS expert researchers in Japan (Prof. Yoshinobu, Tokyo Univ. with research staffs at the endstation of BL07LSU beamline at Spring-8). We could not find contaminated characterization results in our data when we have maintained strict impurity control protocol for CO₂ gas at the beamline facilities in the last 5 years. The related results and discussions are also found elsewhere (Koitaya et al., *Top. Catal.* 2016, **59**, 526-531; Koitaya et al., *ACS Catal.* 2019, **9**, 4539-4550). We added descriptions of gas feed and cleaning procedure details in Methods on the revised manuscript to help potential readers understand better, as below.

– Page 17: Method - Synchrotron-based NAP-XPS experiments.

“High purity CO (99.999%) and CO₂ (99.995%) gas cylinders were connected to a compact-sized gas manifold, and each streamed gas was further purified by repeated cycles of freeze-pump-thaw using a cold trap. All gas lines had a bake out procedure at 383 K with a high-speed pump out for at least 12 hr before the cleaned gas feed in experiments.”

Reviewer comment #3-3: *Overall the assignments of the C1s XPS are not clearly explained, in my opinion – what is the identity of the so-called CO₂(dis) species in Figure 3? Is this distinct from what you are labelling *CO?*

Response #3-3: We agree that our peak assignments for C 1s NAP-XPS results seem ambiguous. *Bent*-structured molecular geometry can share adsorption sites on the Rh surface at a characterized cross-sectional area by X-ray photon beams because of the unique adsorption feature of CO₂ on the surface at NAP. Unlike the CO adsorption structure, the evolved peaks of assigned chemisorbed CO₂ in C 1s spectra show only average ensembles during effective molecular collisions at the interface. Extensive efforts at vibrational mode investigation and potential energy calculation work (Freund and Roberts, *Surf. Sci. Rep.* 1996, **25**, 225-273.) support our argument of unusual CO₂ interactions, indicating that the bent geometry with elongated C-O bonds has a higher enthalpy of 0.5 eV than linear CO₂ (Pacansky et al., *J. Chem. Phys.* 1975, **62**, 2740.; Compton et al., *J. Chem. Phys.* 1975, **63**, 3821.). Their subdivided discrete configurations of ground and excited states of CO₂ were well established by Hatree-Fock

calculations (Winter et al., *Chem. Phys. Lett.* 1973, **20**, 489-492.). This picture of physical meaning suggests that possibly indistinct CO₂ adsorption happens to be on the Rh(111) surface with segmented formation energies at the touch of molecules within 90 μs (Cooper and Compton, *J. Chem. Phys.* 1973, **59**, 3550-3565.); the unstable CO₂⁻ metastable structure was also experimentally observed by an electron attachments process, due to the poor Franck–Condon overlap at the surface energy potential of ground and excited states. We carefully assigned each evolved peak in C 1s NAP-XPS spectra to explain the CO₂ adsorption features, but it is still challenging work to determine the meaning of each peak clearly. Because our NAP-XPS setup does not support time-resolved characterizations below the time scale of μs, we can only observe the suspicious features during effective molecular collisions between CO₂ and the Rh(111) surface. Of course, we know that the *t*-CO and *h*-CO species which were well-characterized at 0.1 mbar CO have similar behavior with C2 and C1, as shown in **Figure 3d**, but it does not mean the evolved peaks have same features each other. As proposed, our calculation works on CO₂ dissociation over Rh(111), and the “CO”-O can directly contact Rh atoms at the initial stage; the other side of “O”-CO would share adjacent adsorption sites of Rh surface at transition states. NAP-XPS measurements show these rounds of the CO₂ dissociation process as overlapped chemical binding species, which are assigned as CO₂(ads.) and CO₂(dis.) in our manuscript. According to the reviewer’s comment, we have simplified the peak assignments for the C 1s NAP-XPS analysis during the CO₂ dissociation, as below.

“CO₂(ads.): Adsorbate CO₂ on the Rh(111) surface.

CO(dis.): Dissociated CO from the CO₂ adsorbate. It is characterized as CO* on the NAP-XPS image in **Figure 3a.**”

Reviewer comment #3-4: *It is claimed from the STM data in Figure 1 that there is physisorbed CO₂ (intact) adsorbed on the surface, but from the XPS data in Figure 2, only dissociated CO₂ is assigned, how do the authors reconcile this?*

Response #3-4: We thank the reviewer for their excellent insight on bridging imaging and spectroscopic techniques. In principle, the reviewer’s question is reasonable, and we also considered the existence of characterized *l*-CO₂ species on the NAP-STM

image for NAP-XPS experiments. In our manuscript, **Figure 2b** contains the collected NAP-XP spectra at equilibrium, not an initial stage of peak evolution before CO₂ dissociation. Thus, we can see from the relatively higher blue peak in the CO₂ gas environment in **Figure 2b** that the time-lapse increment portion of the dissociated CO species is explained in **Figure 3d** on our manuscript. Comparably, our NAP-STM image (**Figure 1e**) indicates that the physisorption *l*-CO₂ would likely be observed at an early stage in a time sequence of presented images at 0.1 mbar CO₂. To address this point, we have added evidence of physisorption feature observation in the NAP-XP spectrum, as below.

Figure R3-2. Acquired C 1s NAP-XP spectrum at 0.1 mbar CO₂ after a gas exposure of 8 min.

A peak of CO₂^{δ-} in **Figure R3-2** is one of the generally accepted assignments for weakly-bound physisorption CO₂ species, also identified in previous reports (Eren et al., *J. Am. Chem. Soc.* 2016, **138**, 8207-8211.; Heine et al., *J. Am. Chem. Soc.* 2016, **138**, 13246-13252.). Our peak assignment is well-matched within a line at the binding energy of 288–290 eV according to references.

“However, the effective collisions of CO₂ free molecules on the Rh(111) surface would form weakly bound CO₂(ads.) species^{20,22}, as observed in Figure 1e, which differs remarkably from the measured NAP-XPS results in the CO(g) environment. This unique property is characterized at the early stage of gaseous CO₂ exposure, and the assigned peak of CO₂^{δ-} at 289.4 eV in C 1s core-level spectrum (Supplementary Fig. 8) suggests a clear evidence of molecular CO₂ interactions with Rh atoms at NAP.”

– Supplementary Figure 8.

“Adsorbate-sensitive C 1s NAP-XP spectrum under 0.1 mbar CO₂(g) environment at a different distance of sample-to-aperture. This spectroscopic result was measured after the gas exposure of 8 min at 300 K. The assigned peaks of CO₂(ads.), CO₂^{δ-}, and CO₂(g) are corresponding to chemisorption, physisorption, and scattered gas molecule species. Contrary to Figure 2b, the CO₂(ads.) species for chemisorption is also assigned at 286.1 eV instead of CO(dis.), because the CO₂ dissociation process is not yet fully triggered at that time.”

Reviewer comment #3-5: *What is the origin of the significant difference in gas phase peak heights in Figure 2? The CO peak is much lower in the C1s compared to the CO₂, however in the O 1s they are similar intensities, and the nominal gas pressure is the same.*

Response #3-5: We thank the reviewer for carefully examining our manuscript to point out details of the NAP-XPS data. We understand the reviewer’s concern about surface characterizations at elevated pressures because of the different peak ratio of gas phase and adsorbates in obtained core-level photoelectron spectra. As is well-known in the ambient pressure XPS community, gas phase peak intensity is typically proportional to the backfilled actual gas pressure in a chamber or reaction cell of the analysis system. However, this statement could be valid in a condition of constant sample-to-aperture distance in measurements (Amann et al., *Rev. Sci. Instrum.* 2019, **90**, 103102.). The electron entrance of a NAP-designed XPS hemispherical analyzer consists of a cone

with a hole (few hundred micrometers in diameter) and differential pumping stage; the scattered photo-emitted electrons from the sample by gas molecules enter into the small hole of the hemispherical analyzer. In principle, we could collect different peak intensity ratios of gas phase/adsorbates on the surface, because the distance of sample-to-aperture is not exactly identical for every measurement in our NAP-XPS setup. Of course, this issue could be minimized as much as possible, if we can employ a high-precision and fully-motorized manipulator for sample handling in the analysis chamber. Unfortunately, we manually controlled our sample position in NAP-XPS setup at the endstation of beamline, so each measurement condition had a slight variation in the critical distance between the sample and the analyzer cone. A similar result using the same NAP-XPS setup is found elsewhere (Toyoshima et al., *Chem. Commun.* 2017, **53**, 12657-12660.), which shows the different CO(g) peak intensity ratio between PdAu(111) and Pd(111) surfaces, even though the exposed gas pressure was strictly kept during the NAP-XPS experiments. Many experts in the NAP-XPS setup understand this phenomenon well, and it has been pointed out that our experimental results could cause a misunderstanding on spectral interpretation or the data quality issue to broad readers in *Nature Communications*. To avoid this concern, we have briefly explained the relationship of aperture distance and photoelectron signal intensity in the “Results” section of the revised manuscript, as below.

– Page 9: X-ray spectroscopy analysis of chemical binding energy.

“We note that the peak intensity ratio of gas phase/adsorbates is critically influenced by sample-to-aperture distance, which is irrelevant to the qualitative characterization of adsorbate species in NAP-XP spectra under CO(g) and CO₂(g) environments.”

Reviewer comment #3-6: *Tip-induced changes have been observed in high-pressure STM experiments before, do the authors have any comments regarding their likelihood in these experiments, and does the tip bias play a role in this?*

Response #3-6: We understand the reviewer’s critical view of STM images, as the “tip effect” of STM has been a problem in the surface science community since the 1980s. In principle, the tunneling current flow between the STM tip and the surface of the

conductive matter is described by a relation of bias voltage and potential energy barrier, because the represented wave function of tunneling electrons occupies an underlying empty-level either on the tip or the sample spontaneously, depending on the direction of applied bias in the closed feedback loop. The Wentzel–Kramers–Brillouin (WKB) approximation explains the generated tunneling current flows regarding the Fermi level as a given equation (Jeong Y. Park, “Scanning Tunneling Microscopy”, *Characterization of Materials*, 2012, John Wiley & Sons, Inc.).

$$I = \int_0^{eV} \rho_s(r, E) \cdot \rho_t(r, E - eV) \cdot T(E, eV) dE \quad \text{eq. (1)}$$

(E : Confined energy with respect to Fermi level, $\rho_s(r, E)$ and $\rho_t(r, E)$: the density of state of sample and tip at a certain location r , $T(E, eV)$: tunneling probability for electrons)

The **eq. (1)** could be simplified in boundary conditions (the density of states for tip and sample are constant, and neglect the energy level of biased voltage for the tunneling barrier), the specified terms in **eq. (1)** are reduced in the equation below.

$$I = \rho_s \rho_t V e^{(-A\sqrt{\varphi}z)} \quad \text{eq. (2)}$$

(V : Bias voltage, A : a number of order 1 with units, φ : tunneling barrier height, z : the separating distance between tip and sample)

Thus, we can establish that the inelastic tunneling current flow is a function of z by **eq. (2)** that perturbed situations at elevated gas pressures during the STM measurement may produce artifacts on the recorded topographic scanning image—particularly in the case that gas molecules adsorbed on the STM tip impede the flow of tunneling current by uninterpreted overlapping of density of states. Thus, many efforts have been made to rule out the tip-contamination effect in near-ambient pressure conditions by tip coating or convex control of tip using a purified CO-termination method (Jensen et al., *J. Vac. Sci. Technol. B* 1999, **17**, 1080-1084.; Kolmakov and Goodman, *Catal. Lett.* 2000, **70**, 93-97.; Starr et al., *Top. Catal.* 2005, **36**, 33-41.). Obviously, NAP-STM images can show the formed new structure of surface morphology under environmental condition (Somorjai and Park, *Physics Today*, 2007, **60**, 48-53.). But the scanning tip may have been affected by surrounding gas molecules at the elevated pressure (Laegsgaard et al., *Rev. Sci. Instrum.* 2001, **72**, 3537.). It is well known in electrochemistry communities that CO₂ molecules could be dynamically activated as a

function of bias voltage via electrolysis at the electrode interface (Lichterman et al., *Energy Environ. Sci.* 2015, **8**, 2409-2416.; Skafte et al., *Nat. Energy*, 2019, **4**, 846-855.).

Thus, we carefully performed NAP-STM experiments of CO₂/Rh(111) that repeatedly measured the local position of the surface, and bias voltage was fixed so far as possible to minimize an unexpected involvement of artifacts by tip crush on the recorded topographic image. In our represented NAP-STM images, the detected physisorption or chemisorption CO₂(ads.) molecules on the Rh(111) surface have an ellipsoidal shape in common at 0.1 mbar CO₂. Because our measurement conditions at 300 K cannot distinguish each adsorbate molecular orbital level, such as HOMO or LUMO, probed at liquid helium temperature using the CO-functionalized tip (Hahn et al., *Phys. Rev. Lett.* 2000, **85**, 1914.; Gross et al., *Phys. Rev. Lett.* 2011, **107**, 086101.). The characterized CO₂ molecules of size and height have almost no change (Lateral length: 5.4 ± 0.4 Å; Height: 0.3 ± 0.1 Å) under the different tunneling conditions of the biased sample voltage. Therefore, the tip-induced modifications are not found on NAP-STM images during the direct observation below 1.0 V of sample bias, as shown in **Figure R3-3**.

Figure R3-3. A recorded NAP-STM image in 0.1 mbar CO₂ at different tunneling condition ($V_t = 0.67$; $I_t = 0.16$ nA).

In our present study, we captured the CO₂(ads.) at three different tunneling conditions during NAP-STM observations under 0.1 mbar CO₂ environment. Even though the characterized mobile CO₂(ads.) are still too fast to characterize in the fast-

scanning mode of our tip scanning capability, most of them could be recognized in their adsorbed form on the Rh(111) surface at that moment. We polished descriptions of NAP-STM images and added a supplementary figure, as below.

– **Page 6: Molecular adsorption structure observations of CO and CO₂.**

“In addition, the characterized CO₂(ads.) molecules are consistently observed at different tunneling conditions in NAP-STM measurements, showing that the recorded tunneling images of CO₂-Rh interface analysis results are far from a tip-induced artifact in direct observations (Supplementary Fig. 4).”

– **Supplementary Figure 4.**

“Representative line profile analysis results of the characterized CO₂(ads.) molecule at different tunneling conditions of 0.22, 0.54, and 0.67 V at 0.1 mbar CO₂ environment, which indicates that the recorded NAP-STM images of CO₂(ads.) molecules are irrelevant to the STM tip-induced artifact between 0.22 and 0.67 V of positively biased voltage.”

Reviewer comment #3-7: *In figure 3, the C1s data has been normalised to the C1 peak, what is the justification for this? It would be good to present the unnormalized data in the supporting information to see the evolution of the total carbon species on the surface with time. Overall this work underlines the challenges of the near ambient pressure techniques and the difficulties in attempting to correlate spectroscopic trends with atomically resolved microscopy. It would significantly benefit from some additional*

complementary spectroscopic probes such as RAIRS to aid in the identification of the C-O intermediate species.

Response #3-7: As suggested by the reviewer, we added a supplementary figure of “unnormalized” C 1s core-level NAP-XP spectra at 0.1 mbar CO₂, as below.

Figure R3-4. The recorded peak intensity changes of CO(dis.) in C 1s NAP-XPS measurements and its histogram analysis results. The unnormalized spectra show the gradual increment of dissociated CO from CO₂ over the Rh(111) surface.

As pointed out by the reviewer, we displayed C 1s spectra with normalization to the C1 peak in **Figure 3d** on our manuscript to focus on the increasingly dissociated CO species at that moment. In fact, the C2 also overlapped with the feature of CO₂ adsorption, and we can confirm the two representative peaks at 285.5 and 286.1 eV from an early stage of CO₂ dissociation, due to effective molecular collisions of CO₂. Thus, C1 at 285.5 eV should be kept constantly, so far as backfilled CO₂ pressure is maintained in analysis chamber. As shown in **Figure R3-4**, although the relative intensity of CO₂(ads.) (magenta color) slightly decreased within an exposure time of 60 min, there is no clear trend of chemical reaction after that time. In contrast, the assigned CO(dis.) peak also noticeably increased at 108 min. It is also important, when we interpret the

surface peak of NAP-XPS results, to carefully treat the adsorbate of reactants, because gas molecules around the cone aperture of analyzer have gradient concentrations, depending on the distance of sample-to-aperture (Calderón et al., *J. Chem. Phys.* 2016, **144**, 044706.), which has also been well established at the beamline facility at PF-KEK (Yoshida and Kondoh, *Chem. Rec.* 2014, **14**, 806-818.). Thus, we assumed that our spectral normalization procedure of C 1s core-level spectra at 0.1 mbar CO₂ with respect to peak C1 does not cause a misinterpretation. By the reviewer's suggestion, we additionally provided the "unnormalized" C 1s core-level spectra as a supplementary figure, and the description of **Figure 3** was polished in the revised manuscript as below.

– Figure 3.

"All spectral interpretation procedures were identically carried out before and after normalization of C 1s core-level NAP-XP spectra (Supplementary Fig. 13)."

– Supplementary Figure 13.

"The recorded peak intensity changes of CO(dis.) and CO₂(ads.) in C 1s core-level NAP-XPS measurements and its histogram analysis results. The unnormalized spectra show the gradual increment of dissociated CO from CO₂ over Rh(111) surface, but the measured peak intensities of CO₂(ads.) in time-lapse signal collections are unrelated to the trend of CO(dis.) as a function of lapsed time. Excess amounts of CO₂(g) were backfilling in the analysis chamber when the C 1s core-level spectrum was acquired at each shot of X-ray photon beam irradiation."

We appreciate the reviewer's overall evaluation of our CO₂ dissociation study combined with NAP-STM, NAP-XPS, and DFT calculations. We believe our experimental results could bring attention to the questionable adsorption structure of CO₂ and the details of the dissociation pathway over the Rh(111) surface at NAP in the readership of *Nature Communications*. In our best knowledge, our real-space observation of CO₂ adsorbates catalytic interactions is revealed at NAP for the first time that the characterized key evidence of dissociated CO evolution is also confirmed by employing with synchrotron-based NAP-XPS technique. Moreover, the supported DFT calculation results propose a reasonable reaction pathway of *bent*-structured CO₂

dissociation with a lowered activation energy barrier at transition state, rather than the similar pathway by the *linear*-structured CO₂ over the Rh(111) model catalyst.

Nevertheless, we could not completely reveal the correlation between the CO₂ dissociation process and detailed intermediate formations on the surface at NAP. As we mentioned in **Response #3-3**, our spectroscopic analysis results show limited chemisorption features in CO₂ adsorbate, owing to the short lifetime of excited CO₂ anions. Even though this is a promising indication for opening new insights into CO₂ chemistry at ambient pressure, investigating discrete energy states of CO₂ on metal catalysts at NAP using modern surface science techniques remains a significant challenge.

The RAIRS technique suggested by the reviewer may be an alternative to characterize the intermediate C-O from CO₂ dissociation at NAP, because the identified vibrational mode would produce important information to explicate the detailed process of catalytic reaction steps at a given moment. However, the physical meaning of each specified vibrational mode also requires interpretation, depending on the structural sensitivity. One study by Kim et al., (*ACS Catal.* 2016, **6**, 1037-1044.) is an excellent example of various activated species interpretations of the CO₂ reduction process on the stepped Cu surface. The authors reported carefully investigated vibrational frequencies for $\delta(\text{OCO})$, $\nu_s(\text{OCO})$, $\nu_{as}(\text{OCO})$, $\nu(\text{CH})$, and $\nu(\text{OCO}) + \delta_{ip}(\text{CH})$, corresponding to carboxylate, carbonate, bicarbonate, and formate species, when the backfilled H₂ (750 mTorr) and CO₂ (250 mTorr) mixed gas react on the Cu surface at elevated temperature. We may establish the complexities of vibrational frequencies of CO₂ interactions, which would deliver important evidence of intermediate co-adsorption and hydrogen-induced CO₂ dissociation.

Employing the RAIRS technique to acquire more characterized experimental evidence for our study would doubtless enhance our manuscript. However, the combination of vibrational frequency characterization is beyond our present work on gas-solid interface analysis. Unfortunately, with the worldwide pandemic conditions due to COVID-19, active international collaborations are currently extremely limited; we would like to have a chance of further study using the RAIRS technique in the future.

REVIEWERS' COMMENTS

Reviewer #1 (Remarks to the Author):

I have already mentioned in my previous review, that the research in the present manuscript represents cutting edge science. The questions and comments I asked have been fully and satisfactorily answered. The corresponding changes in the manuscript are excellent and further improve the quality of the manuscript. I recommend publication of the revised MS as it stands.

Sincerely,
Joachim Paier

Reviewer #3 (Remarks to the Author):

The authors have addressed my concerns as highlighted in my initial review.

In particular:

- o The issue for potential x-ray beam-induced changes have been carefully considered, although these are challenging to completely eliminate from such experiments, it appears that to the furthest extent possible, these are not playing a major role in the observed changes.
- o The precautions that the authors took to avoid contamination of the gas feed appear to be of a good standard, however it would be good to see a survey XPS spectrum to confirm the lack of any trace contaminants that may affect the reactivity.

My other comments have been adequately addressed in the revised manuscript.

Dr. David Grinter